# Southeast Atmosphere Studies: learning from model-observation syntheses

Jingqiu Mao[1], Annmarie Carlton[2,*], Ronald C. Cohen[3], William H. Brune[4], Steven S. Brown[5,6], Glenn M. Wolfe[7,8], Jose L. Jimenez[5], Havala O. T. Pye[9], Nga Lee Ng[10], Lu Xu[11], V. Faye McNeill[12], Kostas Tsigaridis[13,14], Brian McDonald[6], Carsten Warneke[6], Alex Guenther[15], Matthew J. Alvarado[16], Joost de Gouw[5], Loretta J. Mickley[17], Eric M. Leibensperger[18], Rohit Mathur[9], Christopher G. Nolte[9], Robert Portmann[6], Nadine Unger[19], Larry W. Horowitz[20]

[1]Geophysical Institute and Department of Chemistry, University of Alaska Fairbanks, Fairbanks, Alaska, USA
[2]Dept. of Environmental Sciences, Rutgers University, New Brunswick, New Jersey, USA
* now at Dept. of Chemistry, Univ. of California, Irvine, CA, USA
[3]Department of Earth and Planetary Science, University of California, Berkeley, California, USA
[4]Department of Meteorology, Pennsylvania State University, University Park, Pennsylvania, USA
[5]Department of Chemistry and CIRES, University of Colorado, Boulder, Colorado, USA
[6]Earth System Research Laboratory, National Oceanic and Atmospheric Administration, Boulder, Colorado, USA
[7]Atmospheric Chemistry and Dynamics Lab, NASA Goddard Space Flight Center, Greenbelt, MD, USA
[8]Joint Center for Earth Systems Technology, University of Maryland Baltimore County, Baltimore, MD, USA
[9]National Exposure Research Laboratory, U.S. Environmental Protection Agency, Research Triangle Park, North Carolina, USA
[10]School of Chemical & Biomolecular Engineering & School of Earth & Atmospheric Sciences, Georgia Institute of Technology, Atlanta, Georgia, USA
[11]Division of Geological and Planetary Sciences, California Institute of Technology, Pasadena, CA, USA
[12]Department of Chemical Engineering, Columbia University, New York, NY USA
[13]Center for Climate Systems Research, Columbia University, New York, NY, USA
[14]NASA Goddard Institute for Space Studies, New York, NY, USA
[15]Department of Earth System Science, University of California, Irvine, California, USA
[16]Atmospheric and Environmental Research, Lexington, Massachusetts, USA
[17]School of Engineering and Applied Sciences, Harvard University, Cambridge, Massachusetts, USA
[18]Center for Earth and Environmental Science, SUNY Plattsburgh, Plattsburgh, NY, USA
[19]College of Engineering, Mathematics and Physical Sciences, University of Exeter, UK
[20]Geophysical Fluid Dynamics Laboratory/National Oceanic and Atmospheric Administration, Princeton, New Jersey, USA

# Abstract

Concentrations of atmospheric trace species in the United States have changed dramatically over the past several decades in response to pollution control strategies, shifts in domestic energy policy and economics, and economic development (and resulting emission changes) elsewhere in the world. Reliable projections of the future atmosphere require models to not only accurately describe current atmospheric concentrations; but to do so by representing chemical, physical and biological processes with conceptual and quantitative fidelity. Only through incorporation of the processes controlling emissions and chemical mechanisms that represent the key transformations among reactive molecules can models reliably project the impacts of future policy, energy, and climate scenarios. Efforts to properly identify and implement the fundamental and controlling mechanisms in atmospheric models benefit from intensive observation periods (IOPs), during which co-located measurements of diverse, speciated chemicals in both the gas and condensed phases are obtained. The Southeast Atmosphere Studies (SAS, including SENEX, SOAS, NOMADSS and SEAC4RS) conducted during the summer of 2013, provided an unprecedented opportunity for the atmospheric modeling community to come together to evaluate, diagnose, and improve the representation of fundamental climate and air quality processes in models of varying temporal and spatial scales.

This paper is aimed to discuss progress in evaluating, diagnosing, and improving air quality and climate modeling using comparisons to SAS observations as a guide to thinking about improvements to mechanisms and parameterizations in models. The effort focused primarily on model representation of fundamental atmospheric processes that are essential to the formation of ozone, secondary organic aerosols (SOA) and other trace species in the troposphere, with the ultimate goal of understanding the radiative impacts of these species in the Southeast and elsewhere. Here we address questions surrounding four key themes: gas phase chemistry, aerosol chemistry, regional climate and chemistry interactions, and natural and anthropogenic emissions. We expect this review to serve as a guidance for future modeling efforts.

# 1. Introduction

The Southeast US has been studied extensively because it includes both intense emissions of biogenic VOC and has multiple large sources of anthropogenic emissions (e.g. Chameides et al., 1988; Trainer et al., 1987). An improved understanding of ozone photochemistry in this region has subsequently led to effective ozone control strategies (Council, 1991). In 1990s, a number of aircraft and ground field campaigns were conducted to study ozone photochemistry in the Southeast US (Cowling et al., 2000, 1998; McNider et al., 1998; Hübler et al., 1998; Meagher et al., 1998; Martinez et al., 2003; Roberts et al., 2002; Stroud et al., 2001). Aggressive regulatory efforts over the past decade have substantially decreased $NO_x$ in this region (e.g. Russell et al., 2012). This decrease is changing the factors that control the $NO_x$ lifetime and offers an opportunity to study mechanisms of emission from ecosystems in the region in different chemical regimes. The decrease in $NO_x$ is also shifting the regime of $HO_x$ chemistry from one where the primary reaction partner for $HO_2$ and $RO_2$ was NO to one where isomerization, $RO_2 + HO_2$ and $HO_2 + HO_2$ are more important. The Southeast Atmosphere Studies (SAS, including SENEX, SOAS, NOMADSS and SEAC4RS), was designed to study the atmospheric chemistry of the region in the context of changing anthropogenic emissions.

Observational experiments in the Southeastern U.S. during SAS (Southeast Atmosphere Studies) 2013 (SOAS, SENEX, SEAC4RS, NOMADSS) provide a wealth of new insights into the composition of the atmosphere. Results allow researchers to explore the chemical degradation of

biogenic organic molecules over a range of concentrations of ambient nitrogen oxide
concentrations during day and night, and the ensuing consequences for ozone, aerosol and radiative
properties of the atmosphere. The experiment was large and collaborative, and included
coordinated measurements at multiple surface sites and, among several aircraft, with many
flyovers of the surface sites and a wide suite of available remote sensing from space based
instruments. A comprehensive array of instruments at each site/aircraft tracked most of the key
atmospheric observables. Direct tracking of oxidative pathways was made possible by including
gas phase measurements of parent molecules and many of the first- and second-generation
daughter molecules. For the first time, many of the daughter molecules were also tracked into the
aerosol phase. These observations provided an important context for both the characterization of
new instruments and new methods by interpreting measurements from more established
instruments. In parallel with these field measurements, several laboratory experiments used the
same instrumentation to provide insights into the chemical mechanisms of oxidation and
instrument performance under field conditions. Overviews of the entire project and many of the
subprojects have been presented elsewhere (Carlton et al., 2017; Warneke et al., 2016; Toon et al.,
2016). Analyses of the observations have ranged from those that focus on the observations alone
to those that primarily describe model simulations of the region. In this review we focus on the
intersection of these two approaches, which is on analyses of observations that specifically test
and inform the construction of 3-D chemical weather models. Our evaluations are focused on the
Southeast data set, although we assert that the lessons learned are global.

## 2. Gas-phase Chemistry

### 2.1 Background

Global and regional models tend to significantly overestimate summertime surface ozone over the
Southeastern US (Fiore et al., 2009; Murazaki and Hess, 2006; Yu et al., 2010; Yu et al., 2007;
Lin et al., 2008; Rasmussen et al., 2012), posing a challenge for air quality management in this
region and elsewhere. It remains unclear whether this model bias in summertime surface ozone is
mainly due to the chemical processes (e.g. $HO_x$ recycling, isoprene nitrate chemistry,
heterogeneous reactions, nighttime chemistry), physical processes (e.g. dry deposition, boundary
layer processes) or emissions. Fiore et al. (2005) suggested that this problem might be due to
incorrect representation of isoprene sources and chemistry. Measured deposition rates for isoprene
oxidation products appear to be higher than current model values (Nguyen et al., 2015a; Karl et
al., 2010). In the meantime, the understanding of isoprene oxidation chemistry has been evolving
rapidly in the past decade (Crounse et al., 2011; Peeters et al., 2014; Peeters et al., 2009), as a
result conclusions drawn from models using older chemical mechanism may not be correct.
A large debate surrounds our understanding of hydroxyl radical (OH) and hydroperoxy radical
($HO_2$) concentrations in the presence of isoprene. Traditional mechanisms assume that isoprene
oxidation suppresses OH concentrations in low-$NO_x$ conditions via the formation of organic
hydroxyperoxides (Jacob and Wofsy, 1988). However, observations show higher-than-expected
OH concentrations in isoprene-rich environments without corresponding enhancements in $HO_2$ or
$RO_2$ (Tan et al., 2001; Carslaw et al., 2001; Lelieveld et al., 2008; Hofzumahaus et al., 2009; Ren
et al., 2008; Pugh et al., 2010; Thornton et al., 2002; Stone et al., 2010), suggesting a gap in current
understanding of isoprene oxidation. On the other hand, an interference has been discovered to
affect some of these OH instruments (Mao et al., 2012; Novelli et al., 2014; Feiner et al., 2016).
Measurements of higher than expected OH in the presence of isoprene spurred renewed interest in
issues related to the products of the $HO_2 + RO_2$ reactions. Thornton et al. (2002) and Hasson et al.
(2004) had pointed out that if this reaction does not terminate the radical chain it would change
the behavior of $HO_x$ radicals at low $NO_x$. Several specific case of the $HO_2 + RO_2$ reactions were
shown to have an OH product (Hasson et al., 2004; Jenkin et al., 2007; Dillon and Crowley, 2008).
Peeters et al. (2009; 2014) identified a new path for OH regeneration through unimolecular
isomerization of isoprene hydroxyperoxy radicals. This pathway was confirmed by laboratory
measurements of its rate (Crounse et al., 2011; Teng et al., 2017). A key feature of the SAS
experiments was that the $NO_x$ concentrations spanned a range that resulted in measurements where
the three major fates of isoprene peroxy radicals (reaction with NO, $HO_2$ or isomerization) were
sampled at different times and locations.
Another major consequence of isoprene oxidation is the production of isoprene nitrates, formed
from $RO_2+NO$ reaction in the isoprene degradation chain during daytime and by addition of $NO_3$
to the double bonds in isoprene or isoprene daughters at night. Different treatments of these
reactions in models including the yield and subsequent fate of daytime isoprene nitrates, cause as
much as 20% variations in global ozone production rate and ozone burden among different models
(Ito et al., 2009; Horowitz et al., 2007; Perring et al., 2009a; Wu et al., 2007; Fiore et al., 2005;
Paulot et al., 2012). Large variations mainly stem from different yield of isoprene nitrates (Wu et
al., 2007) and the $NO_x$ recycling ratio of these isoprene nitrates (Ito et al., 2009; Paulot et al., 2012).
Recent laboratory data indicates the yield of first generation isoprene nitrates is in the range of 9%
to 14% (Giacopelli et al., 2005; Patchen et al., 2007; Paulot et al., 2009a; Lockwood et al., 2010;
Sprengnether et al., 2002; Xiong et al., 2015; Teng et al., 2015), which is much higher than the 4%
that was in favor as recently as 2007 (Horowitz et al., 2007). The subsequent fate of these isoprene
nitrates includes oxidation by OH, $NO_3$ and $O_3$ (Lockwood et al., 2010; Paulot et al., 2009a; Lee
et al., 2014), photolysis (Müller et al., 2014), and hydrolysis. Synthesis of models and SAS
observations suggest an important role for hydrolysis as expected based on the laboratory
measurements (Romer et al., 2016; Fisher et al., 2016; Wolfe et al., 2015).
The SAS observations also provide measurements that guide our thinking about the role of $NO_3$
chemistry and its representation in models, especially as it contributes to oxidation of biogenic
volatile organic compounds (BVOC) at night (Warneke et al., 2004; Brown et al., 2009; Aldener
et al., 2006; Ng et al., 2008; Ng et al., 2017; Edwards et al., 2017). During SAS, these reactions
were a substantial sink of $NO_x$ in addition to their role in oxidation of BVOC. To a large extent
this is due to the high yield of carbonyl nitrates (65%-85%) from the isoprene + $NO_3$ oxidation
(Perring et al., 2009b; Rollins et al., 2009; Rollins et al., 2012; Kwan et al., 2012; Schwantes et al.,
2015). Models that incorporate this chemistry (Xie et al., 2013; Horowitz et al., 2007; von
Kuhlmann et al., 2004; Mao et al., 2013), indicate that the isoprene+$NO_3$ reaction contributes more
than 50% of the total isoprene nitrate production and that the reaction is thus a major pathway for
nighttime $NO_x$ removal. The fate of products from isoprene+$NO_3$ and to what extent they return
$NO_x$ remains a subject of discussion and thus an opportunity for exploration with models that
might guide our thinking about a plausible range of product molecules (Perring et al., 2009b;
Müller et al., 2014; Schwantes et al., 2015).
Compared to isoprene, the oxidation mechanism of monoterpene has received much less attention
partly due to lack of laboratory and field data. In contrast to isoprene, terpene emissions are
temperature sensitive but not light sensitive (Guenther et al., 1995), leading to a significant portion
of terpenes emissions being released at night. Browne et al. (2014) showed that monoterpene
oxidation is a major sink of $NO_x$ in the Arctic and by implication most of the remote atmosphere.
The high yield of organic nitrates and the low vapor pressure and high solubility of monoterpene
organic nitrates results in strong coupling of gas phase mechanisms to predictions of SOA in a
model. For example, the reaction of terpenes+$NO_3$ provides a large source of SOA as inferred (Ng
et al., 2017). These aerosol organic nitrates can be either a permanent or temporary $NO_x$ sink
depending on their precursors as well as ambient humidity (Nah et al., 2016b; Boyd et al., 2015;
Lee et al., 2016a; Romer et al., 2016). Monoterpene organic nitrates can also be formed from
monoterpene oxidation by OH and $O_3$ in the presence of $NO_x$ and some of them may be susceptible
to rapid hydrolysis/photolysis in aerosol phase (thus not detected as aerosol nitrates), leading to an
underestimate of its contribution to SOA mass (Rindelaub et al., 2015; Rindelaub et al., 2016).
Results from ambient field studies show that particulate organic nitrates can contribute 5-77% (by
mass) of submicrometer organic aerosols, depending on the sampling sites and seasons (Ng et al.,
186 2017).

**2.2 Major relevant findings**
A major focus of the SAS study was to study the daytime and nighttime oxidative chemistry of
isoprene and to compare the observations against models representing the ideas outlined above.
Over the range of the fate of the isoprene $RO_2$ radical, isomerization was important and the reaction
partners were mostly NO and $HO_2$ during the day and a mix of $NO_3$, $RO_2$ and $HO_2$ at night. The
field measurements were closely partnered with laboratory chamber experiments (Nguyen et al.,
2014b) which enhanced our understanding of oxidation mechanisms and provided increased
confidence in our understanding of the measurements of isoprene oxidation products. We
summarize these major relevant findings here:
(1) Radical production: Combining traditional laser-induced fluorescence with a chemical removal
method that mitigates potential OH measurement artifacts, Feiner et al. (2016) found that their
tower-based measurements of OH and $HO_2$ during SOAS show no evidence for dramatically
higher OH than current chemistry predicts in an environment with high BVOCs and low NOx.
Instead, they are consistent with the most up-to-date isoprene chemical mechanism. Romer et al.
(2016) found that the lifetime of $NO_x$ was consistent with these OH observations and that the major
source of $HNO_3$ was isoprene nitrate hydrolysis. Their conclusions would be inconsistent with
dramatically higher OH levels, which would imply much more rapid isoprene nitrate production
than observed. Other ratios of parent and daughter molecules and chemical lifetimes are also
sensitive to OH and these should be explored for additional confirmation or refutation of ideas
about OH production at low $NO_x$.
Isoprene vertical flux divergence in the atmospheric boundary layer over the SOAS site and similar
forest locations was quantified by Kaser et al. (2015) during the NSF/NCAR C-130 aircraft flights
and used to estimate daytime boundary layer average OH concentrations of 2.8 to
$6.6 \times 10^6$ molecules $cm^{-3}$. These values, which are based on chemical budget closure, agree to within
20% of directly-observed OH on the same aircraft. After accounting for the impact of chemical
segregation, Kaser et al. (2015) found that current chemistry schemes can adequately predict OH
concentrations in high isoprene regimes. This is also consistent with the comparison between
measured and modeled OH reactivity on a ground site during SOAS, which show excellent
agreement above the canopy of an isoprene-dominated forest (Kaiser et al., 2016).
(2) Isoprene oxidation mechanism: Recent refinements in our understanding of the early
generations of isoprene degradation have stemmed from a synergy of laboratory, field, and
modeling efforts. Laboratory work has provided constraints on the production and fate of a wide
range of intermediates and end products, including organic nitrates (Teng et al., 2015; Xiong et al.,
2015; Lee et al., 2014; Müller et al., 2014), the isoprene $RO_2$ (Teng et al., 2017), IEPOX (St. Clair
et al., 2015; Bates et al., 2014; Bates et al., 2016), MVK (Praske et al., 2015), and MACR (Crounse
et al., 2012). These experiments have been guided and/or corroborated by analyses of field
observations of total and speciated alkyl nitrates (Romer et al., 2016; Nguyen et al., 2015a; Xiong
et al., 2015; Lee et al., 2016a), IEPOX/ISOPOOH (Nguyen et al., 2015a), glyoxal (Min et al.,
2016), HCHO (Wolfe et al., 2016), OH reactivity (Kaiser et al., 2016), and airborne fluxes (Wolfe
et al., 2015). Recent modeling studies have incorporated these mechanisms to some extent and
showed success on reproducing temporal and spatial variations of these compounds (Su et al., 2016;
Fisher et al., 2016; Travis et al., 2016; Zhu et al., 2016; Li et al., 2017; Li et al., 2016), as
summarized in Table 1. Continued efforts are needed to reduce newfound mechanistic complexity
for inclusion in regional and global models.
(3) Oxidized VOC: Large uncertainties remain on the production of smaller oxidation products.
Several modeling studies indicate an underestimate of HCHO from isoprene oxidation in current
mechanisms (Wolfe et al., 2016; Li et al., 2016; Marvin et al., 2017). Current chemical mechanisms
differ greatly on the yield of glyoxal from isoprene oxidation (Li et al., 2016; Chan Miller et al.,
2017). The observations indicate that the ratio of glyoxal to HCHO is 2%, independent of $NO_x$
(Kaiser et al., 2015), and this ratio is reproduced, at least to some extent, in two modeling studies
(Li et al., 2016; Chan Miller et al., 2017). Confirmation of such a ratio is a useful indicator as these
molecules are also measured from space and both are short- lived and tightly coupled to oxidation
chemistry. Widespread ambient confirmation of the ratio is difficult because of large biases in
satellite glyoxal quantification (Chan Miller et al., 2017).
For the case of the major daughter products methylvinylketone (MVK) and methacrolein (MACR),
lab experiments have confirmed that ambient measurements reported to be MVK and MACR, by
instruments with metal inlets including gas chromatography (GC) and proton transfer reaction–
mass spectrometry (PTR-MS), are more accurately thought of as a sum of MVK, MACR and
isoprene hydroperoxides that react on metal and are converted to MVK and MACR (Rivera‐Rios
et al., 2014; Liu et al., 2013).
(4) Organic Nitrates: The assumed lifetime and subsequent fate of organic nitrates can profoundly
influence $NO_x$ levels across urban-rural gradients (Browne and Cohen, 2012; Mao et al., 2013),
affecting oxidant levels and formation of secondary organic aerosol (SOA). Field observations
during SAS suggest a short (2-3 hr) lifetime of total and isoprene/terpene organic nitrates (Wolfe
et al., 2015; Romer et al., 2016; Fisher et al., 2016; Lee et al., 2016a). One possible explanation is
aerosol uptake of these organic nitrates followed by rapid hydrolysis as confirmed in laboratory
experiments (Hu et al., 2011; Darer et al., 2011; Rindelaub et al., 2016; Rindelaub et al., 2015;
Jacobs et al., 2014; Bean and Hildebrandt Ruiz, 2016), although the hydrolysis rate varies greatly
with the structure of nitrate and aerosol acidity (Hu et al., 2011; Rindelaub et al., 2016; Boyd et
al., 2017; Boyd et al., 2015).
(5) Nighttime Chemistry: The SAS studies examined nighttime BVOC oxidation in both the
nocturnal boundary layer (NBL) and the residual layer (RL). Measurements at the SOAS ground
site provided a wealth of detailed information on nighttime oxidation processes in the NBL via
state of the art instrumentation to constrain the major oxidants, BVOCs and gas and aerosol phase
products (Ayres et al., 2015; Xu et al., 2015b; Lee et al., 2016a). A major focus of these efforts
was to understand the influence of nitrate radical ($NO_3$) oxidation as a source of secondary organic
aerosol.  These results are reviewed in Section 3.2.3 below, and show that organic nitrates from
reactions of $NO_3$ with monoterpenes are an important SOA source in the NBL.  Reactions of
monoterpenes dominate nighttime chemistry near the surface due to their temperature (but not
sunlight) dependent emissions and their accumulation to higher concentration in the relatively
shallow NBL.
Nighttime flights of the NOAA P-3 probed the composition of the overlying residual layer and the
rates of nighttime oxidation processes there.  In contrast to the NBL, isoprene dominates the
composition of BVOCs in the RL, with mixing ratios over Alabama on one research flight
demonstrating a nighttime average near 1 ppbv.  Monoterpene mixing ratios were more than an an
order of magnitude lower. Consumption of isoprene by $O_3$ and $NO_3$ was shown to depend on the
sunset ratio of $NO_x$ to isoprene, with $NO_3$ reaction dominating at ratios above approximately 0.5
and $O_3$ reaction dominant at lower ratios.  Overall, $O_3$ and $NO_3$ contributed approximately equally
to RL isoprene oxidation in the 2013 study.  This observation, combined with recent trends in $NO_x$
emissions, suggests that RL nighttime chemistry in the southeast U.S. is currently in transition
from a $NO_x$ dominated past to an $O_3$ dominated future, a condition more representative of the pre-
industrial past.  The implications of this trend for understanding organic nitrates and secondary
organic aerosol should be considered in models of the influence of changing $NO_x$ emissions on
BVOC oxidation (Edwards et al., 2017).
(6) HONO: The community's confusion about sources of HONO was not resolved by SAS.
Airborne observations over water from the NCAR C130 suggest that conversion of $HNO_3$ to
HONO and $NO_x$ via photolysis of particulate nitrate in the marine boundary layer is important (Ye
et al., 2016). A separate study using NOAA WP-3D observations indicates that HONO mixing
ratios in the background terrestrial boundary layer are consistent with established photochemistry
(Neuman et al., 2016). Persistent uncertainties regarding the potential for measurement artifacts
continue to hamper efforts to resolve outstanding questions about putative novel HONO sources.
(7) Higher-order terpenes: Monoterpene and sesquiterpene chemistry requires continued
investigation. Initial studies indicate that monoterpene oxidation can be an important sink of $NO_x$
and an important source of aerosol precursors (Lee et al., 2016a; Ayres et al., 2015). Additional
analysis is needed to understand the role of monoterpenes. We note that because our understanding
of isoprene chemistry has been changing so rapidly and because the role of isoprene sets the stage
for evaluating the role of monoterpenes, we are now in a much better position to evaluate the role
of monoterpene chemistry.

**2.3 Model recommendations**
Based upon the improved understanding outlined above, we make the following recommendations
for the future modeling efforts:
(1) Measurements and modeling effort on OH show no indication of a need for empirical tuning
factors to represent OH chemistry in the rural Southeast US. Detailed mechanisms based on recent
laboratory chamber studies (mostly at Caltech) and theoretical studies (Leuven) for isoprene result
in predicted OH that is in reasonable agreement with observations (Figure 1). Condensed
mechanisms that approximate the detailed ones are expected to do the same. Whatever mechanism
is used, a key diagnostic identified are parent-daughter molecular relationships such as $NO_2/HNO_3$
or MVK/isoprene. Models calculations should emphasize opportunities for observations of such
ratios as an independent measure of the effect of OH on the atmosphere.
(2) The chemistry of isoprene should be treated in more detail than most other molecules. We
recommend that there should be explicit chemistry through the first and second generation of
isoprene oxidation. No other species should be lumped with isoprene or its daughters. Even for
climate models that cannot afford this level of complexity, a reduced mechanism of isoprene
oxidation should be generated for a wide range of conditions.
(3) $NO_3$ chemistry is an important element of both VOC oxidation and aerosol production and
should be included in models that do not explicitly take it into account, both as a loss process of
VOCs and as a source of aerosols.
(4) The largest $NO_x$ and BVOC emissions are not collocated, as one is mainly from mobile sources
and power plants and the other one is mainly from forests (Yu et al., 2016; Travis et al., 2016). As
a result, model resolution can impact predicted concentrations of trace species. Different model
resolutions may lead to as much as 15% differences at the tails of the $NO_x$ and HCHO
distribution—less so for $O_3$ (Yu et al., 2016; Valin et al., 2016). Depending on the research
question models should evaluate the need to resolve this last 15% which requires a horizontal
resolution of order 12 km or less.
**2.4 Key model diagnostics**
We identified a number of key diagnostics that should probably be evaluated before a model is
used to pursue more interesting new questions. These include:
(1) $NO_x$ concentrations from *in situ* and satellite observations. Models that do not predict the
correct magnitude of $NO_x$ should produce the wrong OH, $O_3$, and parent:daughter VOC ratios (e.g.
Isoprene: Isoprene + IEPOX, Isoprene : MACR + MVK). At the low $NO_x$ characteristic of the
Southeast U.S. these errors are approximately linear—that is, a 15% error in $NO_x$ should
correspond to a 15% error in OH, isoprene and other related species. Given the difficulty in
predicting $NO_x$ to this tolerance, caution should be taken not to over interpret model predictions.
(2) HCHO from space based observations is emerging as a useful diagnostic of model oxidation
chemistry (Valin et al., 2016).
(3) A significant fraction of isoprene remains at sunset and is available for oxidation via $O_3$ or $NO_3$
at night. Analysis of nighttime isoprene and its oxidation products in the residual layer in the
northeast U.S. in 2004 suggested this fraction to be 20% (Brown et al. 2009). Preliminary analysis
from SENEX suggested a similar fraction, although the analysis depends on the emission inventory
for isoprene, and would be 10-12% if isoprene emissions were computed from MEGAN (see
Section 4.2 for the difference between BEIS and MEGAN). This fact might be a useful diagnostic
of boundary layer dynamics and nighttime chemistry in models. The overnight fate of this isoprene
depends strongly on available $NO_x$ (see above). More exploration of the model prediction of the
products of $NO_3$ + isoprene and additional observations of those molecules will provide insight
into best practices for using it as a diagnostic of specific model processes.
(4) $O_3$ and aerosol concentrations and trends over decades and contrasts between weekdays and
weekends across the Southeast remain a valuable diagnostic of model performance, especially as
coupled to trends in $NO_x$ on those same time scales.
**2.5 Open questions**
There are many open questions related to gas phase chemistry. Here we highlight a few that we
believe are best addressed by the community of experimentalists and modelers working together
(there were many other open questions that we think could be addressed by individual investigators
pursuing modeling or experiments on their own).
(1) The sources and sinks of $NO_x$ are not well constrained in rural areas that cover most of
Southeast U.S. As anthropogenic combustion related emissions experience further decline, what
do we expect to happen to $NO_x$? What observations would test those predictions?
(2) As we are reaching consensus on a mechanism for isoprene oxidation, the role of monoterpene
and sesquiterpene oxidation is becoming a larger fraction of remaining uncertainty. Strategies for
exploring and establishing oxidation mechanisms for these molecules and for understanding the
level of detail needed in comprehensive and reduced mechanisms are needed.
(3) Water in aerosol (and cloud) is identified as an important control over gas-phase concentrations.
What are the controls over the presence and lifetime of condensed liquid water? What model and
observational diagnostics serve as tests of our understanding?
(4) Air quality modeling efforts have long been most interested in conditions that are not of top
priority to meteorological researchers—e.g. stagnation. In addition to a better understanding of
horizontal flows in stagnant conditions these experiments highlighted the need for a deeper
understanding of the links between chemical mixing and boundary layer dynamics in day and night.
A number of new chemical observations have been identified in the Southeast US data sets.
Combined approaches using models and these observations to guide our thinking about PBL
dynamics are needed.

## 3. Organic aerosol

### 3.1 Background

Improving the representation of organic aerosol (OA) is a critical need for models applied to the
Southeast. Current air quality and chemistry-climate models produce a very wide range of organic
aerosol mass concentrations, with predicted concentrations spread over 1-2 orders-of-magnitude
in free troposphere (Tsigaridis et al., 2014). Secondary OA (SOA) has traditionally been modeled
by partitioning of semivolatile species between the gas and aerosol phase (Odum et al., 1996;
Chung and Seinfeld, 2002; Farina et al., 2010), but very large uncertainties remain on the detailed
formulations implemented in models (Spracklen et al., 2011; Heald et al., 2011; Tsigaridis et al.,
2014). In particular, the recent identification of substantial losses of semivolatile and intermediate
volatility species to Teflon chamber walls (Matsunaga and Ziemann, 2010; Zhang et al., 2014;
Krechmer et al., 2016; Nah et al., 2016a) necessitate a re-evaluation of the gas-phase SOA yields
used in models which has yet to be comprehensively performed. Models have difficulties to
reproduce the mass loading of OA in both urban and rural areas, although order-of-magnitude
underestimates have only been observed consistently for urban pollution (e.g. Volkamer et al.,
2006; Hayes et al., 2015). For example, CMAQ underestimates OA by 17% at SEARCH network
sites with higher overestimates and underestimates at night and during the day respectively (Pye
et al., 2017a). Furthermore, current OA algorithms often rely on highly parameterized empirical
fits to laboratory data that may not capture the role of oxidant (OH vs $O_3$ vs $NO_3$) or peroxy radical
fate. The peroxy radical fate for historical experiments in particular, may be biased compared to
the ambient atmosphere where peroxy radical lifetimes are longer and autoxidation can be
important.
Recent laboratory, field and model studies suggest that a significant fraction of SOA is formed in
aqueous phase cloud droplets and aerosols, following gas-phase oxidation to produce soluble
species (Sorooshian et al., 2007; Fu et al., 2008; Myriokefalitakis et al., 2011; Carlton et al., 2008;
Tan et al., 2012; Ervens et al., 2011; Volkamer et al., 2009). This is also consistent with the strong
correlation between OA and aerosol liquid water in the Southeast US over the past decade (Nguyen
et al., 2015b). A number of gas-phase VOC oxidation products have been recognized as important
precursors for aqueous production of SOA, including epoxides (Pye et al., 2013; Nguyen et al.,
2014a; Surratt et al., 2010) and glyoxal (Liggio et al., 2005; Woo and McNeill, 2015; McNeill et
al., 2012). Aerosol uptake of these oxygenated VOCs can be further complicated by aerosol acidity
and composition (Pye et al., 2013; Paulot et al., 2009b; Nguyen et al., 2014a; Marais et al., 2016;
Sareen et al., 2017).
While a significant portion of ambient OA has been attributed to various source classes and
precursors (e.g. BBOA from biomass burning, IEPOX-SOA from isoprene epoxydiols or IEPOX,
and less-oxidized oxygenated OA, LO-OOA from monoterpenes), a large portion of ambient OA
(e.g. more-oxidized oxygenated OA, MO-OOA) remains unapportioned. This portion lacks
detailed chemical characterization or source attribution, so further investigation is warranted (Xu
et al., 2015b; Xu et al., 2015a). A diversity of modeling approaches, including direct scaling with
emissions, reactive uptake of gaseous species, and gas-aerosol partitioning etc., is encouraged to
provide insight into OA processes, while trying to make use of all available experimental
constraints to evaluate the models.
**3.2 Major relevant findings**
A number of modeling groups will be interested in modeling aerosol for the Southeast Atmosphere
Study (SAS) across a variety of spatial and temporal scales. Different studies will be able to
support different levels of detail appropriate for their application. Detailed box model
representations can serve to confirm or refute mechanisms and, eventually, be condensed for
application at larger scales such as those in chemical transport or global climate models. In the
following sections, we highlight areas of organic aerosol that should be represented.
**3.2.1 Partitioning theory and phases**
No large kinetic limitations to partitioning are observed in the southeast and partitioning according
to vapor pressure is active on short timescales (Lopez-Hilfiker et al., 2016). The higher relative
humidity in this region, which results in fast diffusion in isoprene-SOA containing particles (Song
et al., 2015), may be at least partially responsible for this behavior. In some instances (e.g. for key
IEPOX-SOA species), observations indicate that detected OA species are significantly less volatile
than their structure indicates, likely due to thermal decomposition of their accretion products or
inorganic-organic adducts in instruments (Lopez-Hilfiker et al., 2016; Hu et al., 2016; Isaacman-
VanWertz et al., 2016; Stark et al., 2017).
Further research is needed regarding the role of organic partitioning into OA versus water and this
can be evaluated using field data. If both processes occur in parallel in the atmosphere, vapor
pressure dependent partitioning to OA may occur along with aqueous processing without
significant double counting or duplication in models. However, due to the high relative humidity
(average RH is 74%, see Weber et al. (2016)) and degree of oxygenation of organic compounds
(OM/OC is 1.9-2.25, see below) in the southeast US atmosphere, inorganic-rich and organic-rich
phases may not be distinct (You et al., 2013) and more advanced partitioning algorithms
accounting for a mixed inorganic-organic-water phase may be needed (Pye et al., 2017a; Pye et
al., 2017b).
Phase separation can be predicted based on determining a separation relative humidity (SRH),
which is a function of degree of oxygenation and inorganic constituent identity (You et al., 2013),
and comparing to the ambient relative humidity. For RH<SRH, phase separation occurs. Pye et al.
(2017a), predicted phase separation into organic-rich and electrolyte-rich phases occurs 70% of
the time during SOAS at CTR with a higher frequency during the day due to lower RH.

### 3.2.2 Primary organic aerosol

Primary organic aerosol concentrations are expected to be small in the Southeast outside urban
areas and we make no major recommendation for how to model them. Modelers should be aware
that a fraction of primary organic aerosol (POA) based on the EPA National Emission Inventory
(NEI) is semivolatile (Robinson et al., 2007). However, not all POA is thought to be semivolatile
– for example, OA from sources such as soil are included in the NEI. Modeled POA may already
include some oxidized POA (OPOA) (if the models include heterogeneous oxidation (as in CMAQ
(Simon and Bhave, 2012)), or hydrophilic conversion (as in GEOS-Chem (Park et al., 2003))).
Thus care should be exercised in evaluating model species such as POA with Aerosol Mass
Spectrometer (AMS) Positive Matrix Factorization (PMF) factors such as hydrocarbon-like OA
(HOA). For semivolatile POA treatments, mismatches between POA inventories and
semivolatile/intermediate volatility organic compounds (S/IVOCs) needs to be carefully
considered. Comparisons of model inventory versus ambient ratios of POA/$\Delta$CO, POA/black
carbon (BC), or POA/$NO_x$ can be used to indicate whether or not POA emissions are excessive
(De Gouw and Jimenez, 2009). As these ratios can be affected by errors in the denominator species,
it is important to also evaluate those carefully against observations. For models with limited POA
information, the ratio of organic matter to organic carbon (OM/OC) should be adjusted to reflect
the highly oxidized nature of ambient OA (as mass is transferred from hydrophobic/hydrophilic
concentrations for example). The OM/OC ratio of bulk ambient OA in the Southeast US is 1.9-
2.25 as measured during summer 2013 (Kim et al., 2015; Pye et al., 2017a).
A biomass burning PMF factor (BBOA) was observed during SOAS and likely has a higher impact
on brown carbon (BrC) than its contribution to OA mass would suggest, although overall BrC
concentrations were very small (Washenfelder et al., 2015). Net SOA mass added via
photochemical processing of biomass burning emissions is thought to be modest, relative to the
high POA emissions (Cubison et al., 2011; Jolleys et al., 2012; Shrivastava et al., 2017).

### 3.2.3 Particle-phase organic nitrates

Organic nitrates, primarily from monoterpene reactions with the nitrate radical, have been
recognized as an important source of OA in the southeast, contributing from 5 to 12% in Southeast
US in summer (Xu et al., 2015a; Ayres et al., 2015; Pye et al., 2015; Xu et al., 2015b; Lee et al.,
2016a). In fact, this number could be an underestimate if some of these organic nitrates are
susceptible to hydrolysis or photodegradation, and thus are not detected as nitrates. We have high
confidence that models should include SOA formation from nitrate radical oxidation of
monoterpenes. Sesquiterpenes and isoprene may also contribute OA through nitrate radical
oxidation, but the contribution is expected to be smaller (Pye et al., 2015; Fisher et al., 2016). A
number of options exist for representing this type of aerosol including fixed yields, Odum 2-
product parameterizations, volatility basis set (VBS) representations (Boyd et al., 2015), and
explicit partitioning/uptake of organic nitrates (Pye et al., 2015; Fisher et al., 2016).
Detailed modeling studies can provide additional insight into the interactions between
monoterpene nitrate SOA and gas-phase chemistry, as well as the fates of specific organic nitrates.
Explicit formation and treatment of organic nitrates, yields of which are parent hydrocarbon
specific, can take into account hydrolysis of particle-phase organic nitrate (ON). The hydrolysis
should depend on the relative amounts of primary, secondary, and tertiary nitrates which are
produced in different abundances in photooxidation vs. nitrate radical oxidation (Boyd et al., 2015;
Boyd et al., 2017). Hydrolysis may also depend on the level of acidity and presence of double
bonds in the organic nitrate (Jacobs et al., 2014; Rindelaub et al., 2016). In addition to hydrolysis,
particle organic nitrates could photolyze and release $NO_x$ or serve as a $NO_x$ sink through deposition
(Nah et al., 2016b).
Formation of organic nitrates should also be considered in the context of emerging evidence for
the role of autoxidation, especially in the monoterpene system (Ehn et al., 2014). Autoxidation has
been shown to occur in both photooxidation and ozonolysis of monoterpenes (Jokinen et al., 2015)
and leads to highly oxidized species including organic nitrates (Lee et al., 2016a; Nah et al., 2016b),
many of which are low volatility. While some empirical representations (e.g. VBS or Odum 2-
product) of monoterpene SOA may capture these species, autoxidation products may be very
susceptible to chamber wall loss (Zhang et al., 2014; Krechmer et al., 2016) and missing from
SOA parameterizations. The role of autoxidation in forming SOA in the southeastern US
atmosphere remains to be determined. In this regard, future laboratory studies should carefully
constrain the peroxy radical reaction channels (e.g. Schwantes et al., 2015; Boyd et al., 2015) and
be conducted under regimes that are representative of ambient environments where the peroxy
radical lifetimes can vary.

### 3.2.4 Isoprene epoxydiol (IEPOX) SOA
Due to the abundance of observations in the Southeastern atmosphere (Budisulistiorini et al., 2016;
Hu et al., 2015b; Xu et al., 2015a; Xu et al., 2015b; Xu et al., 2016; Hu et al., 2016), similarity
between laboratory and field IEPOX-SOA determined by PMF analysis, and availability of model
parameterizations to predict IEPOX-SOA (Pye et al., 2013; Woo and McNeill, 2015; Marais et al.,
2016; Budisulistiorini et al., 2017), we have high confidence that IEPOX-SOA should be included
in models. D'Ambro et al. (2017) predicts IEPOX will be the major precursor to SOA under low-
$NO_x$ conditions when peroxy radical lifetimes are atmospherically relevant, which has not always
been the case in older experiments. However, a number of parameters needed to predict IEPOX-
SOA are uncertain and different modeling approaches, as well as the use of all available
experimental constraints, could be beneficial. The mechanism of IEPOX-SOA formation involves
gas-phase reactions followed by aqueous processing which can occur either in aerosols or cloud
droplets, although the acid-catalyzed initiation step of the epoxide ring opening favors SE USA
aerosol conditions and makes this process less efficient in cloud water. This mechanism could be
represented as heterogeneous reaction with a reactive uptake coefficient or more explicit
partitioning and particle reaction (Table 1).
The correlation of IEPOX-SOA with sulfate (Xu et al., 2015a; Xu et al., 2016; Hu et al., 2015b)
can serve as a useful model evaluation technique as underestimates in sulfate could lead to
underestimates in IEPOX-SOA in models (Figure 2). Current pathways for IEPOX-SOA
formation (Eddingsaas et al., 2010) involve acidity in aqueous solutions (Kuwata et al., 2015), but
several studies suggest that IEPOX-SOA is not correlated well with aerosol acidity or aerosol
water (Budisulistiorini et al., 2017; Xu et al., 2015a). Ion balances or other simple measures of
aerosol acidity are likely inadequate to characterize particle acidity and thermodynamic models
such as ISORROPIA II or AIM are more appropriate for modeling IEPOX-SOA (Guo et al., 2015;
Weber et al., 2016). Currently, different observational datasets indicate different nominal ratios of
ammonium to sulfate (Pye et al., 2017b), so it needs to be kept in mind that some measurements
report only inorganic sulfate (e.g. ion chromatography) while others report total (inorganic +
organic) sulfate (e.g. AMS). A modeling study suggested that ammonia uptake might be limited
by organics, thus affecting acidity (Kim et al., 2015; Silvern et al., 2017).
SAS observations also provide estimates of some components of IEPOX-SOA including 2-
methyltetrols and IEPOX-organosulfates (Budisulistiorini et al., 2015; Hu et al., 2015b). For
modeling applications focusing on IEPOX-SOA, additional speciation of IEPOX-SOA (into
tetrols, organosulfates, etc.) and oligomerization and volatility can be treated. Treating the
monomers (e.g. 2-methyltetrols) explicitly with their molecular properties will likely lead to
excessive volatility of the IEPOX-SOA (Lopez-Hilfiker et al., 2016; Hu et al., 2016; Isaacman-
VanWertz et al., 2016; Stark et al., 2017).
**3.2.5 Glyoxal SOA**
New information on glyoxal SOA is emerging in this area but its importance in the Southeast
remains unclear. Glyoxal has been suspected to be the dominant aqueous SOA source under high-
$NO_x$ ($RO_2$ + NO) oxidation conditions (McNeill et al., 2012) and the Southeast has a mix of high-
$NO_x$ and low-$NO_x$ ($RO_2$ + $HO_2$) conditions (Travis et al., 2016). In addition, abundant isoprene
emissions can lead to substantial glyoxal concentrations. Modeling for the southeastern U.S.
indicates significant SOA can form from glyoxal (Marais et al., 2016; Pye et al., 2015; Knote et
al., 2014; Li et al., 2016; Chan Miller et al., 2017). Implementation in models may require
modifications to the gas-phase chemistry to specifically track glyoxal which may be lumped with
other aldehydes (e.g. in CB05). Recent model studies do not find that a large SOA source from
glyoxal is required to match observations, but more field measurements and laboratory studies,
especially of the yield from isoprene oxidation and the aerosol uptake coefficient, are required to
constrain the process.
**3.2.6 Cloud SOA**
Results from SOAS and SEAC4RS indicate only a modest enhancement of OA due to cloud
processing over the SE US, which was not statistically significant (Wagner et al., 2015). In addition,
epoxide reactions in cloud droplets are predicted to lead to minor amounts of SOA due to the pH
dependence of IEPOX hydrolysis (Fahey et al., 2017; McNeill, 2015).
**3.2.7 SOA from Anthropogenic Emissions**
While the rural southeast is assumed to be dominated by SOA from biogenic precursors (which
may be influenced by anthropogenic pollution) as a result of high modern carbon (Hidy et al.,
2014), SOA from anthropogenic VOCs is known to play a role from fossil carbon measurements
(~18% at Centerville) (Kim et al., 2015), but it is not directly apportioned otherwise. We note that
since ~50% of urban POA and 30% of urban SOA is non-fossil (Zotter et al., 2014; Hayes et al.,
2015), an urban fraction of ~28% for the SOAS site is consistent with observations (Kim et al.,
2015). This source is as large as most of the other individual sources discussed in this section, and
should not be neglected in modeling studies. A simple parameterization based on CO emissions
(Hayes et al., 2015) may be adequate for incorporating this source in modeling studies and has
shown good results for the Southeast US (Kim et al., 2015), but care should be taken to evaluate
the CO emissions when using it.
**3.2.8 Surface network observations of organic aerosols**
We list several caveats for the process of comparing model results to surface network observations.
OC measurements from IMPROVE surface sites may be biased low in the summer due to
evaporation of organic aerosols during the sample collection and handling (Kim et al., 2015). On
the other hand, SEARCH measurements agree well with research community instruments in
Centerville site, such as AMS. Therefore the SEARCH data should be considered as the reference.
Decreases in mass concentrations of particulate sulfate and nitrate over the past decades is
consistent with environmental policy targeting their gas phase precursors, namely SOx and NOx
emissions. Reductions in particulate organic carbon in the southeastern U.S. over the past decade
(Blanchard et al., 2016; Blanchard et al., 2013) are more difficult to reconcile because in the
summertime it is predominantly modern and there is no control policy aimed at reducing biogenic
VOCs. Decreased SOx (Kim et al., 2015; Xu et al., 2015b; Blanchard et al., 2013) and NOx
emissions modulate the amount of organic aerosol formation through the gas phase impacts
described above, and impacts on the absorbing medium amount (Nguyen et al., 2015b; Attwood
et al., 2014) and chemical composition.
In addition to sources and sinks of OA, attention should also be paid to the role of dry deposition
of gases in determining mass loadings, as this process can have a large impact on model predictions
and is very poorly constrained (Glasius and Goldstein, 2016; Knote et al., 2015).
**3.2.8 Climate relevant properties**
A motivating goal of the southeast studies was to examine PM mass measurements at the surface
and satellite-measured AOD, to facilitate improved prediction of the total aerosol loading. Aerosol
mass aloft contributes to AOD (Wagner et al., 2015), and this complicates the relationship to
surface concentrations. Relative humidity, vertical structure of the daytime PBL, and aerosol liquid
water (not measured by surface networks) influences remotely sensed AOD (Brock et al., 2016a;
Brock et al., 2016b; Kim et al., 2015; Nguyen et al., 2016). AOD is also complicated by aerosol
composition. Attwood et al. (2014) finds that the steeper decrease in sulfate aerosol relative to
organic from 2001 to 2013, has changed the hygroscopicity of SE US aerosol, leading to lower
aerosol liquid water and thus lower optical extinction and AOD.
**3.3 Model recommendations**
Based upon the improved understanding outlined above, we make the following recommendations
for the future modeling efforts:
(1) There is high confidence that a pathway of SOA formation from isoprene epoxydiol (IEPOX)
should be included in models. However, since many of the parameters needed to predict IEPOX-
SOA are uncertain, further mechanistic studies are needed to address these uncertainties.
(2) There is high confidence that models should include SOA formation from nitrate radical
oxidation of monoterpenes (with or without explicit nitrate functionality). Sesquiterpenes and
isoprene may also contribute SOA through nitrate radical oxidation, but the contribution is
expected to be smaller.
(3) More field measurements and laboratory studies, especially of the yield from isoprene
oxidation and the aerosol uptake coefficient, are required to constrain the importance of glyoxal
SOA.
(4) There is high confidence that models should predict SOA from urban emissions with a
parameterization that results in realistic concentrations. The non-fossil fraction of urban POA and
SOA needs to be taken into account when interpreting modern carbon measurements.
(5) Current SOA modeling efforts should be coupled with an up-to-date gas-phase chemistry, to
provide realistic concentrations for several important SOA precursors, including IEPOX, glyoxal,
organic nitrates etc.
**3.4 Open questions**
A number of open questions remain that would benefit from modeling studies:
(1) What is the role of particle-phase organic nitrates in removing or recycling $NO_x$ from the
system?
(2) How much detail do models need to represent in terms of types of organic nitrate (ON)?
(3) What are the formation mechanisms of highly oxygenated organics?
(4) What anthropogenic sources of SOA are models missing?
(5) What climate-relevant aerosol properties are needed in models?
(6) What is the role of clouds in forming and processing organic aerosols?
# 4. Emissions
**4.1 Background**
Emission inventories are a critical input to atmospheric models, and reliable inventories are needed
to design cost-effective strategies that control air pollution. For example, in the 1970s and 1980s,
emission control strategies implemented under the Clean Air Act emphasized the control of
anthropogenic VOC emissions over $NO_x$ (National Research Council, 2004). Despite large order
of magnitude reductions in anthropogenic VOC emissions (Warneke et al., 2012), abatement of
$O_3$ was slow in many regions of the country. In the late 1980s, a large and underrepresented source
of biogenic VOC emissions was identified (Trainer et al., 1987; Abelson, 1988; Chameides et al.,
1988), putting into question the effectiveness of anthropogenic VOC emission control strategies
to mitigate $O_3$ nationally (Hagerman et al., 1997). Since the mid-1990s, large reductions in $NO_x$
emissions have resulted from: (i) controls implemented at power plants (Frost et al., 2006), (ii)
more durable three-way catalytic converters installed on gasoline vehicles (Bishop and Stedman,
2008), and (iii) more effective regulation of diesel $NO_x$ emissions from heavy-duty trucks
(Yanowitz et al., 2000; McDonald et al., 2012). Emission reductions implemented on combustion
sources, have also been linked to decreases in organic aerosol concentrations observed in both
California (McDonald et al., 2015) and the Southeastern U.S. (Blanchard et al., 2016). Though
substantial progress has been made in improving scientific understanding of the major biogenic
and anthropogenic sources of emissions contributing to air quality problems, some issues remain
in current U.S. inventories and are highlighted below.
The Southeast US is a region that has both large natural emissions and anthropogenic emissions.
The accurate knowledge of biogenic emissions is key to understanding many of the processes that
lead to ozone and aerosol formation. Previous studies suggest that MEGANv2.1 can estimate twice
as large isoprene emissions compared with BEIS over the Eastern US (Warneke et al., 2010;
Carlton and Baker, 2011), but most global models using MEGANv2.1 do not show a significant
bias of isoprene over the Southeast US (Mao et al., 2013; Millet et al., 2006). This is likely due to
different landcover data being used in the regional and global applications of MEGAN. Validation
of the various biogenic emission inventories was therefore one of the main science questions for
the SAS studies.

The National Emissions Inventory (NEI) developed by U.S. EPA, is an inventory of air pollutants released every three years, and commonly used in U.S.-based air quality modeling studies. A recent modeling study reported that $NO_x$ emissions from mobile source emissions were overestimated by 51-70% in the Baltimore-Washington, D.C. region (Anderson et al., 2014). Past studies have also found discrepancies in motor vehicle emission models used by EPA to inform the NEI (Parrish, 2006; McDonald et al., 2012). Additionally, problems have been identified in estimates of $NO_x$, VOC, and methane emissions from U.S. oil and gas development (Ahmadov et al., 2015; Pétron et al., 2014; Brandt et al., 2014). Some major oil and gas basins of note are located in the Southeastern U.S., which were measured by aircraft during the SAS2013 studies. In contrast to mobile source and oil and gas emissions, power plant emissions of $NO_x$ and $SO_x$ are believed to be known with greater certainty since large stationary sources of emissions are continuously monitored. In addition to biogenic emission inventories, the datasets collected by the SAS2013 studies have provided an opportunity to assess the accuracy of anthropogenic emissions and their impacts on atmospheric chemistry.

The topic of model resolution, which involves the relationship between emissions and chemistry, is also key to interpreting model-observation comparisons. Regional-scale air quality models can be simulated at very high horizontal resolutions (e.g., 1 km and finer) (Joe et al., 2014); however, typically they are run at coarser resolutions, such as at 12 km by 12 km (e.g., continental U.S.)(Gan et al., 2016) or 4 km by 4 km (e.g., urban scale) (Kim et al., 2016b). The horizontal resolution of global chemistry models has significantly improved, with nesting being performed at horizontal resolutions as fine as 0.25°x0.3125° degree (Travis et al., 2016). Coarse model resolutions can complicate evaluations with high spatial and temporal-resolution measurements (e.g., from aircraft) of chemical constituents undergoing fast chemistry (e.g., isoprene, OH) (Kaser et al., 2015). Sharp concentration gradients are observable from space for species with relatively short atmospheric lifetimes (e.g., nitrogen dioxide, formaldehyde, and glyoxal), and potentially provide insights into the role of natural and anthropogenic emissions on air quality (Duncan et al., 2010; Russell et al., 2012; Lei et al., 2014). Lastly, some emission sources are described by large emission intensities (e.g., power plants and biomass burning), which result in elevated concentrations of emitted species downwind. A coarse model will artificially dilute these high emission fluxes (e.g., $NO_x$ and $SO_x$) over a wider area, which could alter the chemical regime by which ozone (Ryerson et al., 1998; Ryerson et al., 2001) and secondary aerosols (Xu et al., 2015a) form.

## 4.2 Major relevant findings

### 4.2.1 Biogenic emissions

Isoprene emissions measured by the NOAA P3, using the mixed boundary layer budget method, and NCAR/NSF C-130 and NASA DC-8 aircraft using direct eddy covariance flux measurements were within the wide range of observations reported by previous studies. The two methods of estimating isoprene emissions agreed within their uncertainties (Yu et al., 2017). Solar radiation and temperature measured by the aircraft along the flight tracks and available from regional model and assimilations (e.g., WRF, NLDAS-2) enabled estimation of emissions using models including BEIS3.12, BEIS3.13, MEGAN2.0, MEGAN2.1 with default landcover, MEGAN2.1 with revised landcover, and MEGAN3. Isoprene emissions are highly sensitive to solar radiation and temperature and biases in the values used to drive emission models can result in errors exceeding 40% and complicating efforts to evaluate biogenic emission models. As has previously been noted in the southeastern US, MEGAN2.1 predicted isoprene emissions in the Southeast US were about twice as high as BEIS3.13. The measurements fall between the two models and are within the

model and measurement uncertainties (Warneke et al. 2010). Isoprene mixing ratios were modeled
with a) WRF-Chem using BEIS and with b) CAMx using MEGAN and the results were consistent
with the measurement-inventory comparison: WRF-Chem was biased low and CAMx biased high
(Warneke et al., in preparation).
Landcover characteristics including Leaf Area Index (LAI) and tree species composition data are
also critical driving variables for BEIS and MEGAN isoprene and monoterpene emission estimates.
Airborne flux measurements agreed well with MEGAN2.1 for landscapes dominated by
southeastern oaks, which are high isoprene emitting tree species, but landscapes that had an
overstory of non-emitters, with the high isoprene emitters in the understory, showed emissions
lower than expected by the model. The isoprene emission factor was linearly correlated with the
high isoprene emitter plant species fraction in the landcover data set. This may indicate a need for
models to include canopy vertical heterogeneity of the isoprene emitting fraction (Yu et al., 2017).
A simplification used in current biogenic emission models including BEIS3.13, BEIS3.6, and
MEGAN2.1 is that all high isoprene emitting species are assigned the same isoprene emission
factor. For example, all North American species of Quercus (oak), Liquidambar (sweetgum),
Nyssa (tupelo), Platanus (sycamore), Salix (willow), Robinia (locust) and Populus (poplar and
aspen) are assigned a single value based on the average of an extensive set of enclosure
measurements conducted in North Carolina, California and Oregon in the 1990s (Geron et al.,
2001). Earlier studies had reported isoprene emission factors for these tree species that ranged
over more than an order of magnitude (Benjamin et al., 1996). Geron et al. (2001) showed that by
following specific measurement protocols, including leaf cuvettes with environmental controls and
ancillary physiological measurements such as photosynthesis, the variability dropped from over
an order of magnitude to about a factor of 3. They concluded that this remaining variability was
due at least as much to growth conditions as to species differences and so recommended that a
single isoprene emission factor be used for all of these species. Recent aircraft flux measurements
(Misztal et al., 2016; Yu et al., 2017) indicate that there is at least a factor of two difference in the
isoprene emission factors of these species. This could be due to a genetic difference in emission
capacity and/or differences in canopy structure. The aircraft measurements indicate that sweetgum
and tupelo emission factors are similar to the value used in BESI3.13 and BEIS3.6 while the
California oak emission factor is similar to that used in MEGAN2.1. The aircraft based estimate
of southeastern oak emission factors falls between the BEIS3.6 and MEGAN2.1 values. As a result,
aircraft flux measurements in the southeastern US are higher than BEIS3.13/BEIS3.6 and lower
than MEGAN2.1. The MEGAN3 emission factor processor provides an approach for synthesizing
available emission factor data and can be used to account for the emission rate variability observed
by these aircraft flux studies (Guenther et al., in preparation).
Modeling monoterpene emissions is even more challenging than isoprene emissions for reasons
that include multiple emission processes (e.g., both light dependent and light independent
emissions), stress-induced emission capability present in many plant species but not always
expressed, and the potential for enclosure measurements to dramatically overestimate emissions
due to release of monoterpenes from damaged storage pools. The eddy covariance flux
measurements on the NCAR/NSF C130 are similar to the values estimated by MEGAN2.1 for
needle leaf forests, considered to be high emission regions, but are higher than the modeled
monoterpene emissions from other landscapes (Yu et al., 2017). They conclude that unaccounted
processes, such as floral and stress emissions, or sources such as non-tree vegetation may be
responsible for the unexpectedly high monoterpene emissions observed by the aircraft.

During the experiment direct observations of fluxes for a variety of species from large aircraft were conducted, enabling a first direct estimate of fluxes over a regional domain (Wolfe et al., 2015; Yuan et al., 2015; Kaser et al., 2015). These data have the potential for enabling analyses of strengths and weaknesses of current emission and deposition schemes and their implementation within chemical transport models. Vertical flux profiles also contain information on the chemical production and loss rates, providing a new observational constraint on the processes controlling reactive gas budgets. An LES model was used to simulate isoprene, $NO_x$ and their variability in the boundary layer. The results showed good agreement between the measurements and the model. The atmospheric variability of isoprene, the altitude profile in the boundary layer of isoprene and $NO_x$ mixing ratios and fluxes were well reproduced in the model, which was used to validate the eddy covariance and mixed boundary layer methods of estimating isoprene fluxes (Kim et al., 2016a; Wolfe et al., 2015).

### 4.2.2 Anthropogenic emissions

Travis et al. (2016) utilizing the GEOS-Chem model report that $NO_x$ emissions are significantly overestimated by the NEI 2011, and suggest that mobile source and industrial emissions of $NO_x$ need to be lowered by 30-60% to be consistent with aircraft measurements collected over the Southeastern U.S. during the SEAC4RS Study. These results are consistent with modeling studies performed during the DISCOVER-AQ field campaign, which also found that the NEI 2011 overestimated $NO_x$ emissions (Anderson et al., 2014; Souri et al., 2016). However, a later study by Li et al. (2017) utilizing the AM3 model during the SENEX Study suggests that overestimates in NEI 2011 $NO_x$ emissions may be smaller than reported in the Travis et al. study (~14% vs. 30-60%). McDonald et al. (in preparation) using WRF-Chem, found mobile source emissions in the NEI 2011 to be overestimated by ~50% and a factor of 2.2 for $NO_x$ and CO, respectively, when evaluated with SENEX aircraft measurements. Due to rapidly declining trends in vehicle emissions (McDonald et al., 2013; McDonald et al., 2012), some of the emissions overestimate was attributed to utilizing a 2011 inventory in 2013 model simulations. However, roadside measurements of vehicular exhaust also suggest systematic overestimates in emission factors used by EPA's vehicle emissions model (MOVES), likely contributing to the consistent reporting to date of overestimated mobile source $NO_x$ emissions (Anderson et al., 2014; Souri et al., 2016; Travis et al., 2016). When $NO_x$ emissions were reduced from mobile sources by this amount, model predictions of $O_3$ over the Southeastern U.S. were improved both for mean concentrations and $O_3$ extreme days (McDonald et al., in preparation), consistent with modeling by Li et al. (2017) demonstrating the sensitivity of $O_3$ to $NO_x$ emissions in the Southeastern U.S. over the 2004-2013 timespan.

Along with other aircraft field campaigns and tall tower measurements in the Upper Midwest, data from the SENEX Study was used to assess anthropogenic emissions of VOCs in the NEI and a global inventory (RETRO). Hu et al. (2015a) found that RETRO consistently overestimates U.S. emissions of C6-C8 aromatic compounds, by factors of 2 - 4.5; the NEI 2008 overestimates toluene by a factor of 3, but is consistent with top-down emission estimates for benzene and C8 aromatics. The study also suggests that East Asian emissions are an increasingly important source of benzene concentrations over the U.S., highlighting the importance of long-range transport on U.S. air quality as domestic sources of emissions decline (Warneke et al., 2012).

Two studies have quantified top-down emissions of oil and gas operations, derived from aircraft measurements for VOCs and methane from SENEX P-3 data (Peischl et al., 2015; Yuan et al., 2015). The oil and gas regions measured during SENEX account for half of the U.S. shale gas production, and loss rates of methane to the atmosphere relative to production were typically lower

than prior assessments (Peischl et al., 2015). Yuan et al. (2015) explored the utility of eddy-
covariance flux measurements on SENEX and NOMADDS aircraft campaigns, and showed that
methane emissions were disproportionately from a subset of higher emitting oil and gas facilities.
Strong correlations were also found between methane and benzene, indicating that VOCs are also
emitted in oil and gas extraction. High wintertime $O_3$ has been found in the Uinta Basin, UT
(Ahmadov et al., 2015; Edwards et al., 2014), though it is unclear at this time how significant oil
and gas emissions of VOCs could be in an isoprene-rich source region on tropospheric $O_3$
formation. Future atmospheric modeling efforts of oil and gas emissions are needed.
During the SENEX and SEAC4RS studies, research aircraft measured agricultural fires over the
Southeast. Liu et al. (2016) reported emission factors of trace gases, which were consistent with
prior literature. In general, the authors' found emissions of $SO_2$, $NO_x$, and CO from agricultural
fires to be small relative to mobile sources (<10%). However, within fire plumes, rapid $O_3$
formation was observed, indicating potential air quality impacts on downwind communities. To
represent the impact of biomass burning, air quality models need improved treatments of initial
VOC and $NO_x$ emissions and near source chemistry. Sub-grid parameterizations, based on detailed
models like the Aerosol Simulation Program (ASP) (Alvarado and Prinn, 2009) and which
incorporates gas-phase chemistry, inorganic and organic aerosol thermodynamics, and evolution
of aerosol size distribution and optical properties, could improve coarse model representations of
chemistry near biomass burning plumes. Zarzana et al. (2017) investigated enhancements of
glyoxal and methylglyoxal relative to CO from agricultural fires, and report that global models
may overestimate biomass burning emissions of glyoxal by a factor of 4. This highlights large
uncertainties and variability in fire emissions, and a need for additional observational constraints
on inventories and models.

## 4.3 Model Recommendations and Future Work

(1) In the Southeast US isoprene emissions are so large that they influence most atmospheric
chemistry processes. Users of model simulations using the different isoprene inventories have to
be aware of the differences. For example, OH and isoprene concentrations are anti-correlated (Kim
et al 2015) and model simulations using BEIS will potentially have higher OH than simulations
using MEGAN and chemistry will proceed at different rates. In addition, modeled products from
isoprene oxidation in the gas and particle phase will be different. Isoprene derived SOA or
secondary CO in the Southeast US can vary by a factor two between the two inventories.
(2) For future work, BEIS3.6 is now available and needs to be evaluated using the methods
described here.
(3) MEGAN3 emission factor processor can be used to synthesize the available emission factor
estimates from SAS and other studies. A beta version of the MEGAN3 emission factor processor
and MEGAN3 model processes is available and should be evaluated.
(4) A revised $NO_x$ emissions inventory is needed to improve air quality models for $O_3$, especially
in the Southeast U.S. where $O_3$ is sensitive to changes in $NO_x$ emissions. Anthropogenic emissions
of $NO_x$ in the NEI 2011 may be overestimated by 14-60% in the Southeastern U.S. during the
SAS2013 study time period (Travis et al., 2016; Li et al., 2017).

# 5. Chemistry-Climate Interactions

## 5.1 Background

Interactions between atmospheric chemistry and climate over the southeastern United States are not well quantified. The dense vegetation and warm temperatures over the Southeast result in large emissions of isoprene and other biogenic species. These emissions, together with anthropogenic emissions, lead to annual mean aerosol optical depths (AODs) of nearly 0.2, with a peak in summer (Goldstein et al., 2009). The climate impacts of US aerosol trends in the Southeast due to changing anthropogenic emissions is under debate (e.g. Leibensperger et al., 2012b, a; Yu et al., 2014; Tosca et al., 2017). Climate change can, in turn, influence surface air quality, but even the sign of the effect is unknown in the Southeast (Weaver et al., 2009). Part of this uncertainty has to do with complexities in the mechanism of isoprene oxidation, the details of which are still emerging from laboratory experiments and field campaigns (Liao et al., 2015; Fisher et al., 2016; Marais et al., 2016). In addition, the influence of day-to-day weather on surface ozone and particulate matter ($PM_{2.5}$) has not been fully quantified, and climate models simulate different regional climate responses. Resolving these uncertainties is important, as climate change in the coming decades may impose a "climate penalty" on surface air quality in the Southeast and elsewhere (Fiore et al., 2015).

## 5.2 Key science issues and recent advances.

We describe recent advances in four areas related to chemistry-climate interactions in the Southeast.

### 5.2.1. Seasonality and trends in aerosol loading in the Southeast

Using satellite data, Goldstein et al. (2009) diagnosed summertime enhancements in AOD of 0.18 over the Southeast, relative to winter, and hypothesized that secondary organic aerosol from biogenic emissions accounts for this enhancement. Goldstein et al. (2009) further estimated a regional surface cooling of -0.4 W m$^{-2}$ in response to annual mean AOD over the Southeast. These findings seemed at first at odds with surface $PM_{2.5}$ measurements, which reveal little seasonal enhancement in summer. Using SEAC4RS measurements and GEOS-Chem, Kim et al. (2015) determined that the relatively flat seasonality in surface $PM_{2.5}$ can be traced to the deeper boundary layer in summer, which dilutes surface concentrations.

In response to emission controls, aerosol loading over the Southeast has declined in recent decades. For example, wet deposition fluxes of sulfate decreased by as much as ~50% from the 1980s to 2010 (Leibensperger et al., 2012b). Over the 2003-2013 time period, surface concentrations of sulfate $PM_{2.5}$ declined by 60%. Organic aerosol (OA) also declined by 60% even though most OA appears to be biogenic and there is no indication of a decrease in anthropogenic sources (Kim et al., 2015). Model results suggest that the observed decline in OA may be tied to the decrease in sulfate, since OA formation from biogenic isoprene depends on aerosol water content and acidity (Marais et al., 2016; Marais et al., 2017). Consistent with these surface trends, 550-nm AOD at AERONET sites across the Southeast has also decreased, with trends of -4.1% a$^{-1}$ from 2001-2013 (Attwood et al., 2014). Xing et al. (2015a) reported a roughly -4% decrease in remotely sensed AOD across the eastern United States, as measured by the Moderate Resolution Imaging and Spectroradiometer (MODIS) on board Terra and Aqua. These large declines could potentially have had a substantial impact on regional climate, both through aerosol-radiation interactions and aerosol-cloud interactions.

**5.2.2. Contribution of aerosol trends to the U.S. "warming hole."**

Even as global mean temperatures rose over the 20th century in response to increasing greenhouse gases, significant cooling occurred over the central and southeastern United States. This cooling, referred to as the U.S. warming hole (Pan et al., 2004), has been quantified in several ways. For example, Figure 3 shows that annual mean temperatures across the Southeast decreased by ~1 °C during the 1930-1990 timeframe (Capparelli et al., 2013). A different temperature metric, the 20-year annual return value for the hot tail of daily maximum temperatures, decreased by 2 °C from 1950 to 2007 (Grotjahn et al., 2016). Over a similar time frame, Portmann et al. (2009) diagnosed declines in maximum daily temperatures in the Southeast of 2-4 °C per decade, with peak declines in May-June, and linked these temperature trends with regions of high climatological precipitation. Since the early 2000s, the cooling trend has appeared to reverse (Meehl et al., 2015).

The causes of the U.S. warming hole are not clear. Most freely running climate models participating in the Coupled Model Intercomparison Project (CMIP5) cannot capture the observed 20th century temperature trends over the Southeast (Knutson et al., 2013; Kumar et al., 2013; Sheffield et al., 2013); this failure likely arises from either model deficiency or natural variability not included in the simulations. Indeed, several studies have argued that naturally occurring oscillations in sea surface temperatures (SSTs) influenced the large-scale cooling in the Southeast (Robinson et al., 2002; Kunkel et al., 2006; Meehl et al., 2012; Weaver, 2013; Mascioli et al., 2017). Kumar et al. (2013), for example, linked the June-July-August indices of the Atlantic Multidecadal Oscillation (AMO) to annual mean temperatures across the eastern U.S. for the 1901-2004 period. Mauget and Cordero (2014), however, pointed out inconsistencies in these two time series, with the AMO index sometimes lagging temperature changes. A recent study has argued that the transition of the Interdecadal Pacific Oscillation (IPO) phase from positive to negative in the late 1990s may have triggered a reversal of the warming hole trend (Meehl et al., 2015).

The cool period in the Southeast coincided with heavy aerosol loading over the region, and several studies have suggested that trends in aerosol forcing may have also played a role in driving the U.S. warming hole. For example, Leibensperger et al. (2012b, 2012a) found that the regional radiative forcing from anthropogenic aerosols led to a strong regional climate response, cooling the central and eastern US by 0.5-1.0 °C from 1970-1990 (Figure 3), with the strongest effects on maximum daytime temperatures in summer and autumn. In that study, the spatial mismatch between maximum aerosol loading and maximum cooling could be partly explained by aerosol outflow cooling the North Atlantic, which strengthened the Bermuda High and increased the flow of moist air into the south-central United States. Another model study diagnosed positive feedbacks between aerosol loading, soil moisture, and low cloud cover that may amplify the local response to aerosol trends (Mickley et al., 2012). The strength of such positive feedbacks may vary regionally, yielding different sensitivities in surface temperature to aerosol forcing.

The cool period in the Southeast coincided with heavy aerosol loading over the region, and several studies have suggested that trends in aerosol forcing may have also played a role in driving the U.S. warming hole. For example, Leibensperger et al. (2012b, 2012a) found that the regional radiative forcing from anthropogenic aerosols led to a strong regional climate response, cooling the central and eastern US by 0.5-1.0 °C from 1970-1990 (Figure 3), with the strongest effects on maximum daytime temperatures in summer and autumn. In that study, the spatial mismatch between maximum aerosol loading and maximum cooling could be partly explained by aerosol outflow cooling the North Atlantic, which strengthened the Bermuda High and increased the flow of moist air into the south-central United States. Another model study diagnosed positive

feedbacks between aerosol loading, soil moisture, and low cloud cover that may amplify the local
response to aerosol trends in the eastern U.S., including the Southeast (Mickley et al., 2012). The
strength of such positive feedbacks may vary regionally, yielding different sensitivities in surface
temperature to aerosol forcing. More recent modelling studies, however, have generated
conflicting results regarding the role of aerosols in driving the warming hole. For example, the
model study of Mascioli et al. (2016) reported little sensitivity in Southeast surface temperatures
to external forcings such as anthropogenic aerosols or even greenhouse gases. In contrast,
Banerjee et al. (2017) found that as much of 50% of the observed 1950-1975 summertime cooling
trend in the Southeast could be explained by increasing aerosols. Examining multi-model output,
Mascioli et al. (2017) concluded that aerosols accounted for just 17% of this cooling trend in
summer. These contrasting model results point to the challenges in modeling climate feedbacks,
such as those involving cloud cover or soil moisture.
These early model studies have been accompanied by more observationally based efforts to link
trends in surface temperature to aerosol loading. A key first step is to determine whether changes
in surface solar radiation are related to changes in aerosol loading. Measurements from the Surface
Radiation network (SURFRAD) reveal increases of $+0.4$ $Wm^{-2}$ $a^{-1}$ in total surface solar radiation
across the East during 1995-2010 (Gan et al., 2014). An attempt to reproduce the trend in total
surface radiation with a regional chemistry-climate model found a reasonable match with
observations over the East when aerosol-radiation interactions were included (Xing et al., 2015a).
Most of the observed increase in surface solar radiation, however, appears due to increasing diffuse
radiation, at odds with the decline in AOD, which should instead increase direct radiation (Gan et
al., 2015; Gan et al., 2014). Using satellite data and assimilated meteorology, Yu et al. (2014)
showed that trends in spatially averaged AOD and cloud optical depth declined over the 2000-
2011 time period over the eastern US, while daily maximum temperatures and shortwave cloud
forcing increased. These opposing trends suggest that aerosol-cloud interactions may have
influenced the observed ~1 °C warming trend in the Southeast over this ten-year time period, with
the decline in anthropogenic aerosols driving a decrease in cloud cover and a rise in surface
temperatures. Yu et al. (2014) confirmed this hypothesis using a chemistry-climate model. In
contrast, the observational study of Tosca et al. (2017), which also relied on satellite AOD, pointed
to aerosol-radiation interactions as the driver of surface temperature trends in the Southeast.
Analysis of ground-based observations in Mississippi, however, found little covariability between
AOD and clear-sky solar radiation at the surface, casting doubt on the importance of aerosol-
radiation interactions in driving the observed cooling in this region(Cusworth et al., 2017).
Continued improvements of $PM_{2.5}$ air quality in the Southeast may further influence regional
climate. Lee et al. (2016b) projected a warming of about $+0.5$ $Wm^{-2}$ over the eastern U.S.,
including the Southeast, over the 2000-2030 timeframe due to anticipated improvements in air
quality and the associated reduction in AOD. Xing et al. (2015b) have pointed out that an
overlooked beneficial effect of aerosol reduction is increased ventilation of surface air, a positive
feedback that leads to further decline in surface $PM_{2.5}$ concentrations. The feedback arises from
changes in the temperature profile, with warmer temperatures at the surface and cooler
temperatures aloft, which together enhance atmospheric instability and ventilation as aerosol
induced cooling is reduced. The feedback may lead to unexpected health benefits of clearing $PM_{2.5}$
pollution (Xing et al., 2016).

### 5.2.3. Influence of meteorology on surface air quality in the Southeast

Pollution episodes in the southeastern United States are correlated with high temperatures, low wind speeds, clear skies, and stagnant weather (Camalier et al., 2007; Jacob and Winner, 2009). The spatial extent of the Bermuda High also plays a role in modulating air quality in the Southeast (Zhu and Liang, 2013).

Fu et al. (2015) used models and observations to examine the sensitivity of August surface ozone in the Southeast to temperature variability during 1988-2011. This study finds that warmer temperatures enhance ozone by increasing biogenic emissions and accelerating photochemical reaction rates. However, variability in ozone advection into the region may also explain much of the variability of surface ozone, with possibly increased advection occurring during the positive phase of the Atlantic Multidecadal Oscillation (AMO). Applying empirical orthogonal functions (EOF) analysis to observed ozone, Shen et al. (2015) determined that the sensitivity of surface ozone in the Southeast can be quantified by the behavior of the west edge of the Bermuda High. Specifically, for those summers when the average position of the west edge is located west of ~85.4° W, a westward shift in the Bermuda High west edge increases ozone in the southeast by 1 ppbv $deg^{-1}$ in longitude. For all summers, a northward shift in the Bermuda High west edge increases ozone over the entire eastern United States by 1-2 ppbv $deg^{-1}$ in latitude.

The influence of meteorology on $PM_{2.5}$ in the Southeast is not well quantified. Tai et al. (2010) found that observed sulfate and OC concentrations increase with increasing temperature across the region due to faster oxidation rates and the association of warm temperatures with stagnation and biogenic and fire emissions. Nitrate $PM_{2.5}$, however, becomes more volatile at higher temperatures and decreases with temperature. Using local meteorology, however, Tai et al. (2010) could explain only about 20-30% of $PM_{2.5}$ daily variability in the Southeast. Both Thishan Dharshana et al. (2010) and Tai et al. (2012b) diagnosed a relatively weak effect of synoptic scale weather systems on $PM_{2.5}$ air quality in the Southeast, especially in the deep South. Shen et al. (2017), however, extended the statistical studies of Tai et al. (2012a, b) by taking into account not just the local influences of meteorology on $PM_{2.5}$ air quality but also the relationships between local $PM_{2.5}$ and meteorological variables in the surrounding region. These authors developed a statistical model that explains 30-50% of $PM_{2.5}$ monthly variability in the Southeast. Shen et al. (2017) further reported that many atmospheric chemistry models may underestimate or even fail to capture the strongly positive sensitivity of monthly mean $PM_{2.5}$ to surface temperature in the eastern United States, including the Southeast, in summer. In GEOS-Chem, this underestimate can be traced to the overly strong tendency of modeled low cloud cover to decrease as temperatures rise (Shen et al., 2017).

### 5.2.4. Effects of future climate change on Southeast air quality

Emissions of U.S. pollution precursors are expected to decline in coming decades (Lamarque et al., 2013; Fiore et al., 2015), which may offset any potential climate penalty. Background ozone, however, may increase due to increasing methane (West et al., 2012). A major challenge in quantifying the future trends in surface air quality is our lack of knowledge in temperature-dependent isoprene emissions and photochemistry (Achakulwisut et al., 2015).

Using a regional chemistry-climate model, Gonzalez-Abraham et al. (2015) found that daily maximum 8 h average (MDA8) ozone concentrations in the Southeast would likely increase by 3-6 ppbv by the 2050s due solely to climate change and land use change. Changes in anthropogenic emissions of ozone precursors such as methane could further enhance MDA8 ozone in the Southeast by 1-2 ppbv. Rieder et al. (2015), however, determined that large areas of the Southeast

would experience little change in surface ozone by the 2050s, but that study neglected the influence
of warming temperatures on biogenic emissions. Shen et al. (2016) developed a statistical model
using extreme value theory to estimate the 2000–2050 changes in ozone episodes across the United
States. Assuming constant anthropogenic emissions at the present level, they found an average
annual increase in ozone episodes of 2.3 d (>75 ppbv) across the United States by the 2050s, but
relatively little change in the Southeast. In fact, a key result of this work is the relative insensitivity
of ozone episodes to temperature in the Southeast. However, Zhang and Wang (2016) have
suggested that warmer and drier conditions in the Southeast future atmosphere could extend the
ozone season, leading to ozone episodes in October.
Model studies differ on the effects of future climate change on $PM_{2.5}$ in the Southeast. Tai et al.
(2012a) and Tai et al. (2012b) analyzed trends in meteorological modes from an ensemble of
climate models and found only modest changes in annual mean $PM_{2.5}$ ($\pm0.4$ μg m$^{-3}$) by the 2050s
in the Southeast, relative to the present-day. Using a single chemistry-climate model, Day and
Pandis (2015) calculated significant increases of $\sim 3.6$ μg m$^{-3}$ in July mean $PM_{2.5}$ along the Gulf
coast by the 2050s and attributed these increases to a combination of decreased rain-out, reduced
ventilation, and increased biogenic emissions.  Building on the statistical model of Tai et al.
(2012a,b), Shen et al. (2017) found that $PM_{2.5}$ concentrations in the Southeast could increase by
0.5-1.0 μg m$^{-3}$ by 2050 on an annual basis, and as much as 2.0-3.0 μg m$^{-3}$ in summer, assuming
anthropogenic emissions remained at present-day levels.  These authors found that the driver for
these increases was rising surface temperature, which influences both biogenic emissions and the
rate of sulfate production.

## 5.3. Open questions
Unresolved issues in chemistry-climate interactions in the Southeast include the following:
1. What is the impact of aerosols on regional climate of the Southeast? What role do feedbacks
play, including feedbacks involving cloud cover, soil moisture, and boundary layer height? Did
land use changes play a role in the Southeast warming hole? How will changing aerosol
composition affect regional climate? Can we reconcile observed trends in insolation and aerosols?
Can we use observed weekly cycles in temperature or precipitation to probe possible aerosol
effects on regional climate (Forster and Solomon, 2003; Bell et al., 2008; Bäumer et al., 2008;
Daniel et al., 2012)?
2. What caused the U.S. warming hole? Is the observed cooling over the Southeast partly due to
natural variability of North Atlantic SSTs? Do aerosol changes induce changes the North Atlantic
SSTs that feedback on the Southeast U.S.? Has the warming hole ended and made the central and
southeastern United States more vulnerable to high temperatures and drought?
3. What limits model skill in simulating the variability of surface pollution in the Southeast? Can
we capture the observed effects of the Bermuda High or the AMO on surface air quality?
4. How will air quality in the Southeast change in the future? Do current model weaknesses in
simulating present-day ozone and $PM_{2.5}$ daily or seasonal variability limit our confidence in future
projections?

## 5.4. Model recommendations
We recommend the following approaches for studies involving chemistry-climate interactions in
the southeastern U.S.
1. Take advantage of findings from the 2013 measurement campaigns.
For aerosol, such findings include information on composition, hygroscopicity, lifetime, aerosol-
cloud interactions, optical properties, and the mechanism of SOA formation. Modelers should also
take advantage of new information on isoprene emission flux and oxidation mechanisms.
2. Link 2013 results with findings from previous measurement campaigns and with long-term in
situ and satellite data.
3. Work to apply best practices, including standard statistical tests, to chemistry-climate studies.
Modelers need to consider the statistical significance of observed trends and perform ensemble
simulations for robust statistics. The auto-correlation of the variables under investigation should
be examined. Comparison of observed trends with samples of internal climate variability from
model control runs, as in (Knutson et al., 2013), may be a useful approach, and modelers should
acknowledge that observations may represent an outlier of unforced variability.
4. Benchmark chemistry-climate models in a way that is useful for chemistry-climate studies.
For the Southeast, modelers should consider testing the following model properties:

1056        (1) Sensitivity of surface air quality to synoptic weather systems, including the westward extent
1057        of Bermuda High and cold front frequency.
1058        (2) Sensitivity of surface air quality to local meteorological variables and isoprene emissions
1059        on a range of temporal scales.
1060        (3) Sensitivity of soil moisture and cloud cover to changing meteorology and the consequences
1061        for regional climate and air quality.

## 6. Summary
The primary purpose of this work is to improve model representation of fundamental processes
over Southeast US. We summarize the modeling recommendations here:
**Gas-phase chemistry** (1) Up-to-date "standard" chemical mechanisms represent OH chemistry
well over the observed range of $NO_x$ concentrations. Detailed mechanisms based on recent
laboratory chamber studies (mostly at Caltech) and theoretical studies (Leuven) for isoprene
chemistry result in predicted OH that is in reasonable agreement with observations. Condensed
mechanisms that approximate these details are expected to do the same. (2) Given the large
emissions and high chemical reactivity of isoprene, its chemistry should be treated fairly explicitly,
including more detail than for most other hydrocarbons. (3) $NO_3$ chemistry contributes
significantly to both VOC oxidation and aerosol production. (4) The regions of peak $NO_x$ and
BVOC emissions are not collocated. As a result, the model resolution can impact the predictions.
**Organic Aerosol** (1) There is high confidence that a pathway of SOA formation from isoprene
epoxydiol (IEPOX) should be included in models. However, since many of the parameters needed
to predict IEPOX-SOA are uncertain, further mechanistic studies are needed to address these
uncertainties. (2) There is high confidence that models should include SOA formation from nitrate
radical oxidation of monoterpenes (with or without explicit nitrate functionality). Sesquiterpenes
and isoprene may also contribute SOA through nitrate radical oxidation, but the contribution is
expected to be smaller. (3) More field measurements and laboratory studies, especially of the yield
from isoprene oxidation and the aerosol uptake coefficient, are required to constrain the
importance of glyoxal SOA. (4) There is high confidence that models should include SOA from
urban emissions with a parameterization that results in realistic concentrations.
**Natural and anthropogenic emissions** (1) Biogenic emissions from BEIS are generally lower,
and those from MEGAN generally higher, than from measurements for all campaigns. (2)
Observations confirm a rapid decrease of ozone precursor emissions over past few decades. Thus,
use of the correct scaling of anthropogenic emissions for a particular year is important for accurate
simulations. (3) National Emissions Inventory (NEI) 2011 likely overestimates $NO_x$ emissions in
the study area from mobile sources that use fuel-based estimates.
**Climate and chemistry interactions** (1) Annual mean temperatures during the 1930-1990
timeframe decreased by ~1°C over the central and southeastern United States. Several studies have
argued that patterns of sea surface temperatures in the North Atlantic may have caused this large-
scale cooling. Trends in aerosol forcing may have also played a role. (2) Pollution episodes in the
southeastern United States are correlated with high temperatures, low wind speeds, clear skies,
and stagnant weather. Surface air quality over Southeast US may be to some extent modulated by
large-scale circulations, such the Bermuda High or Atlantic Multi-decadal Oscillation (AMO).

# Acknowledgement

This work is based on a workshop held in GFDL in 2015, funded by National Science Foundation
Atmospheric Chemistry Program (AGS-1505306). We acknowledge Haofei Yu (University of
Central Florida), Vaishali Naik (NOAA GFDL), Tom Knutson (NOAA GFDL), John Crounse
(Caltech), Paul Wennberg (Caltech), Daniel Jacob (Harvard), Jen Kaiser (Harvard), Luke Valin
(EPA), Petros Vasilakos (Georiga Tech), Michael Tosca (NASA JPL),
Arlene Fiore (Columbia), Nora Mascioli (Columbia), Yiqi Zheng (Yale), Tzung-May Fu (PKU),
Michael Trainer (NOAA ESRL), Siwan Kim (NOAA ESRL), Ravan Ahmadov (NOAA ESRL),
Nick Wagner (NOAA ESRL), Eladio Knipping (EPRI), for their contributions. We also
acknowledge travel supports from US Environmental Protection Agency (EPA), NOAA Climate
Program Office, and the Cooperative Institute for Climate Science (CICS) at Princeton University.
In particular, we would like to thank Princeton and GFDL staff for support on logistics. We would
also like to thank Ann Marie Carlton's group (Thien Khoi Nguyen, Caroline Farkas, Neha Sareen)
and Luke Valin for additional support on meeting logistics.
**Disclaimer**: Although this document has been reviewed by U.S. EPA and approved for publication,
it does not necessarily reflect U.S. EPA's policies or views.

# 7. Glossary of Acronyms

**AM3**: The atmospheric component of the GFDL coupled climate model CM3.
**AMS**: Aerosol Mass Spectrometer
**AMO**: Atlantic Multi-decadal Oscillation
**AOD**: aerosol optical depth
**BBOA**: Biomass burning OA
**BEIS**: Biogenic Emission Inventory System
**BVOC**: Biogenic Volatile Organic Compounds
**CAMx**: Comprehensive Air Quality Model with Extensions
**CEMS**: Continuous emission monitoring systems
**CMAQ**: Community Multi-scale Air Quality Model
**CSN**: Chemical Speciation Monitoring Network
**EF**: Emission Factor
**FIXCIT**: A laboratory experiment focused on isoprene oxidation chemistry and the instruments
we took to the field to understand that chemistry
**HOA**: Hydrocarbon-like OA
**IEPOX**: Isoprene epoxydiol
**IMPROVE**: Interagency Monitoring of Protected Visual Environments visibility monitoring
network
**LAI**: Leaf Area Index
**LES**: Large-eddy simulation
**LO-OOA**: Less-oxidized oxygenated OA
**MACR**: Methacrolein
**MEGAN**: Model of Emissions of Gases and Aerosols from Nature
**MO-OOA**: More-oxidized oxygenated OA
**MVK**: Methyl vinyl ketone
**NEI**: National Emissions Inventory
**NOAA**: National Oceanic and Atmospheric Administration
**NOMADSS**: Nitrogen, Oxidants, Mercury and Aerosol Distributions, Sources and Sinks aircraft
campaign, took place during Jun-Jul 2013 with the NSF/NCAR C-130 aircraft.
**OA**: Organic aerosol
**OC**: Organic carbon
**OM**: Organic matter
**OMI**: Ozone Monitoring Instrument
**PAN**: Peroxy Acetyl Nitrate
**PFT**: Plant Functional Type
**PMF**: Positive Matrix Factorization
**POA**: primary organic aerosol
**RGF**: Ratio of Glyoxal to Formaldehyde
**SAS**: Southeast Atmosphere Studies
**SCIPUFF**: Second Order Closure Integrated Puff Model
**SEAC4RS**: Studies of Emissions, Atmospheric Composition, Clouds and Climate Coupling by
Regional Surveys aircraft campaign, took place during Aug-Sept 2013 with NASA DC-8 and
ER-2 aircraft
**SEARCH**: Southeastern Aerosol Research and Characterization Network
**SENEX**: SouthEast NEXus of air quality and climate campaign
**S/IVOCs**: Semivolatile/intermediate volatility organic compounds
**SOA**: Secondary Organic Aerosols
**SOAS**: the Southern Oxidant and Aerosol Study ground-based campaign, took place during Jun-
Jul 2013 near Brent, Alabama.
**SURFRAD**: Surface Radiation Budget Network
**VBS**: volatility basis set (VBS)
**WRF-Chem**: Weather Research and Forecasting with Chemistry model

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

Table 1 A subset of model evaluations for SAS observations (till 2017)

| Model name | Model-type | References | Targeted species | Major findings |
|---|---|---|---|---|
| F0AM | 0-D | Feiner et al. (2016) | OH, HO$_2$, OH reactivity | Measured and modeled OH agree well. |
| Box model | 0-D | Lee et al. (2016a) | Speciated organic nitrates | Particle-phase organic nitrates are an important component in organic aerosols, but could have a short particle-phase lifetime. |
| F0AM | 0-D | Wolfe et al. (2016) | HCHO | Current models accurately represent early-generation HCHO production from isoprene but under-predict a persistent background HCHO source. |
| F0AM | 0-D | Kaiser et al. (2016) | OH reactivity | Missing OH reactivity is small. |
| F0AM | 0-D | Marvin et al. (2017) | HCHO | Model HCHO-isoprene relationships are mechanism-dependent. Condensed mechanisms (esp. CB6r2) can perform as well as explicit ones with some modifications. |
| ISORROPIA | 0-D | Weber et al. (2016); Guo et al. (2015) | Aerosol Acidity | Submicron aerosols are highly acidic in Southeast US. |
| MXLCH | 1-D | Su et al. (2016) | Isoprene, HCHO, MVK, MACR, organic nitrates, OH, HO2 | Diurnal evolution of O$_3$ is dominated by entrainment. Diurnal evolution of isoprene oxidation products are sensitive to NO:HO$_2$ ratio. |
| GEOS-Chem | 3-D | Fisher et al. (2016) | Organic nitrates | Updated isoprene chemistry, new monoterpene chemistry, and particle uptake of RONO$_2$. RONO2 production accounts for 20% of the net regional NO$_x$ sink in the Southeast in summer. |
| GEOS-Chem | 3-D | Travis et al. (2016) | NO$_x$, ozone | NEI NO$_x$ emissions from mobile and industrial sources reduced by 30–60%.  The model is still biased high by 6- |

| | | | | |
|---|---|---|---|---|
| | | | | 14 ppb relative to observed surface ozone. |
| GEOS-Chem | 3-D | Zhu et al. (2016) | HCHO | GEOS-Chem used as a common intercomparison platform among HCHO aircraft observations and satellite datasets of column HCHO. The model shows no bias against aircraft observations. |
| GEOS-Chem | 3-D | Kim et al. (2015) | Organic and inorganic aerosols | GEOS-Chem used as a common platform to interpret observations of different aerosol variables across the Southeast.   Surface $PM_{2.5}$ shows far less summer-to- winter decrease than AOD. |
| GEOS-Chem | 3-D | Chan Miller et al. (2017) | Glyoxal, HCHO | New chemical mechanism for glyoxal formation from isoprene. Observed glyxal and HCHO over the Southeast are tightly correlated and provide redundant proxies of isoprene emissions. |
| GEOS-Chem | 3-D | Marais et al. (2016) | IEPOX, organic aerosols | New aqueous-phase mechanism for isoprene SOA formation. Reducing $SO_2$ emissions in the model decreases both sulfate and SOA by similar magnitudes. |
| GEOS-Chem | 3-D | Silvern et al. (2017) | Aerosol acidity | Sulfate aerosols may be coated by organic material, preventing $NH_3$ uptake. |
| GFDL AM3 | 3-D | Li et al. (2016) | Glyoxal, HCHO | Gas-phase production of glyoxal from isoprene oxidation represents a large uncertainty in quantifying its contribution to SOA. |
| GFDL AM3 | 3-D | Li et al. (2017) | Organic nitrates, ozone | Reactive oxidized nitrogen species, including NOx, PAN and $HNO_3$ decline proportionally with decreasing NOx emissions in Southeast U. |
| CMAQ | 3-D | Pye et al. (2015) | Terpene nitrates | Monoterpene + $NO_3$ reactions responsible for significant NOx-dependent SOA. |

| | | | | Magnitude of SOA dependent on assumptions regarding hydrolysis. |
|---|---|---|---|---|
| Box model with CMAQ/Simple-GAMMA algorithms | 0-D | Budisulistiorini et al. (2017); Budisulistiorini et al. (2015) | IEPOX, SOA | Sulfate, through its influence on particle size (volume) and rate of particle-phase reaction (acidity), controls IEPOX uptake at LRK. |
| CMAQ | 3-D | Pye et al. (2017a) | Aerosol liquid water, water soluble organic carbon(WSOC) | Aerosol water requires accurate organic aerosol predictions as models considering only water associated with inorganic ions will underestimate aerosol water. Gas-phase WSOC, including IEPOX+glyoxal+methylglyoxal, is abundant in models. |
| CMAQ | 3-D | Fahey et al. (2017) | Cloud-mediated organic aerosol | Cloud-processing of IEPOX increased cloud-mediated SOA by a modest amount (11 to 18% at the surface in the eastern US) |
| CMAQ | 3-D | Murphy et al. (2017) | Organic aerosol from combustions sources | CTR organic aerosol predictions are not very sensitive to assumptions (volatility, oxidation) for combustion-derived organic aerosol. |
| CMAQ | 3-D | Baker and Woody (2017) | Ozone, PM2.5 | Single-source impacts of a coal fired power plant, including the contribution to secondary pollutants, can be estimated from a 3-D CTM. |
| AIOMFAC, CMAQ | 0-D/3-D | Pye et al. (2017b) | Inorganic aerosol, semivolatile species | Thermodynamic models are consistent with SEARCH and MARGA measured ammonium sulfate at CTR. Organic-inorganic interactions can cause small decreases in acidity and increased partitioning to the particle for organic species with O:C>0.6. |
| WRF-Chem | 3-D | McDonald et al. (in preparation) | $NO_x$, CO, Ozone | Mobile source $NO_x$ and CO emissions overestimated by 50% and factor of 2.2, respectively. Model surface $O_3$ |

| | | | | improves with reduced mobile source $NO_x$ emissions. |
|---|---|---|---|---|
| NCAR LES | 3-D | Kim et al. (2016) | Isoprene, OH | Turbulence impacts isoprene-OH reactivity, and effect depends on $NO_x$ abundance. |



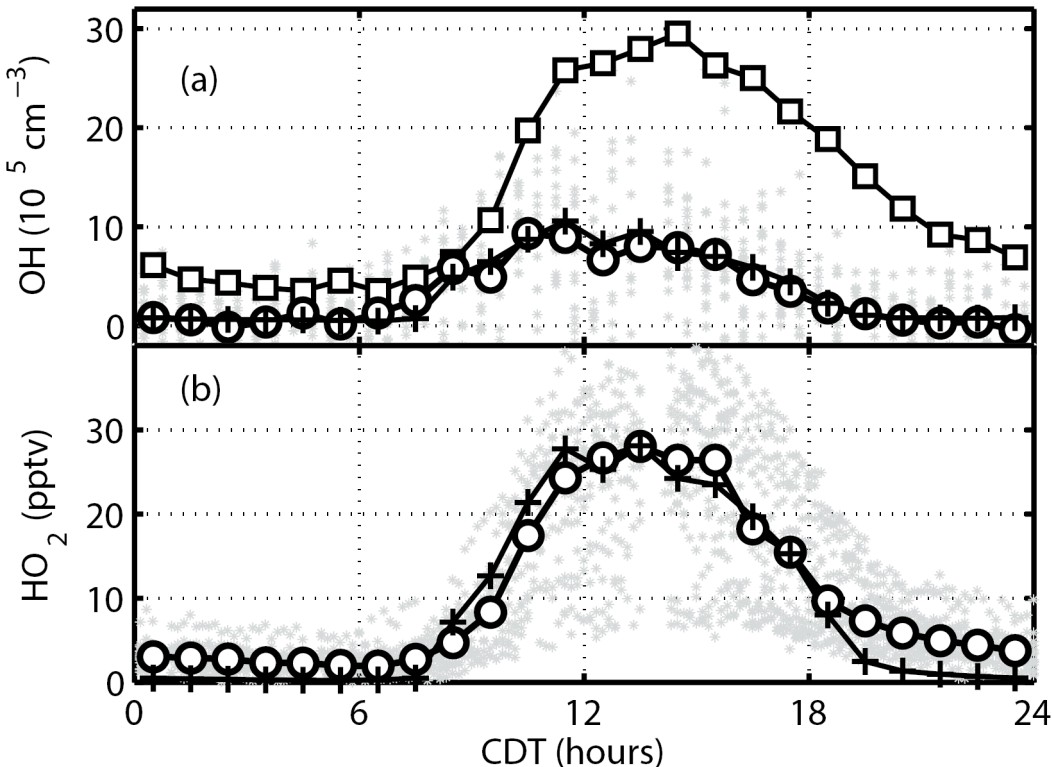

Figure 1 Diel variation of measured and modeled OH/HO$_2$ during SOAS (Feiner et al., 2016). In
panel (a), measured OH by a traditional laser induced fluorescence technique is shown in squares
and by a new chemical scavenger method is shown in circles. The latter one is considered as the
"true" ambient OH. Simulated OH from a photochemical box model with Master Chemical
Mechanism (MCM) v3.3.1 is shown in pluses. In panel (b), measured HO$_2$ is shown in circles and
modeled HO$_2$ is shown in pluses. For both panels, gray dots are individual 10-minute
measurements.


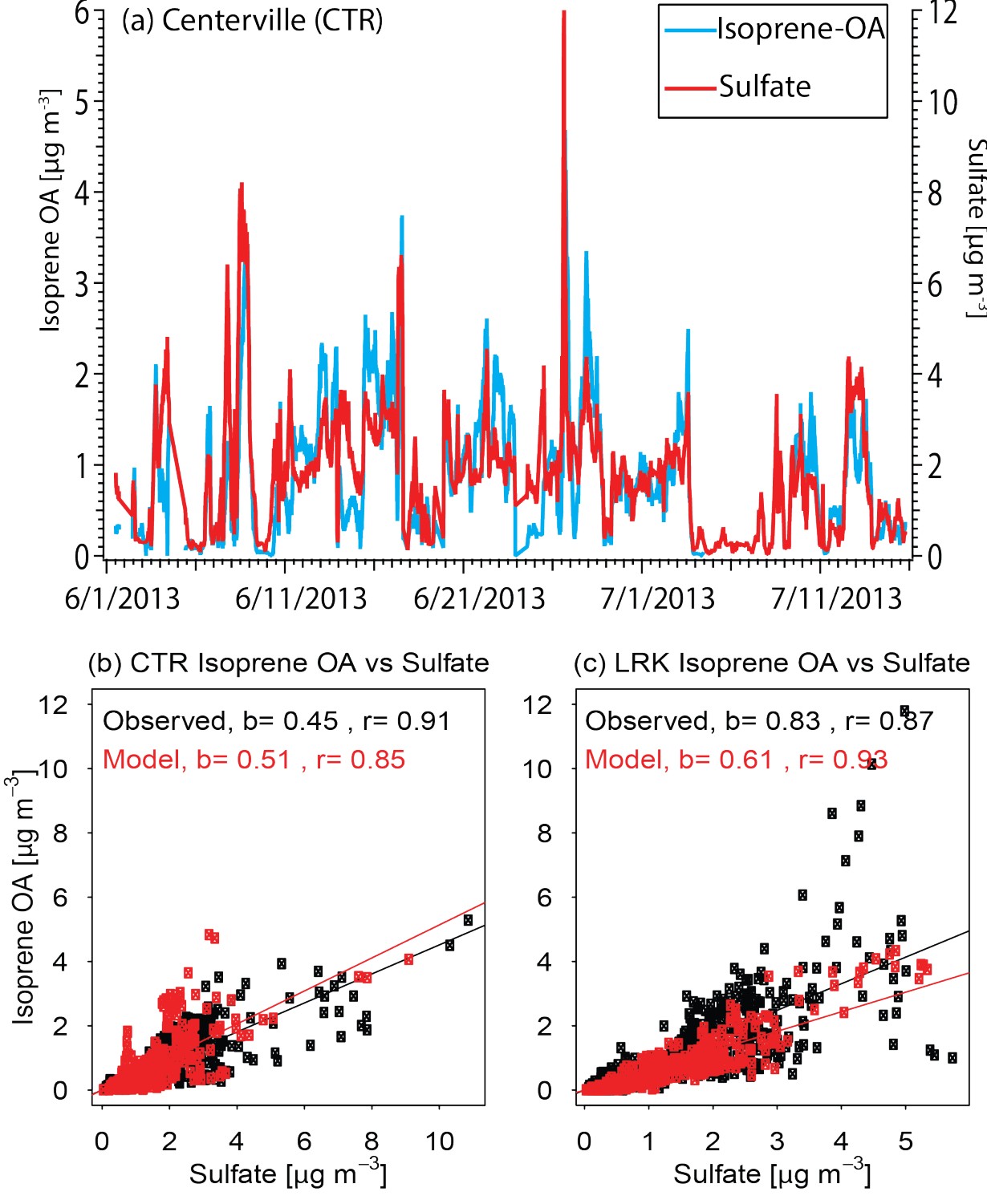

Figure 2. Time series and correlation between Isoprene OA and sulfate during SOAS (Pye et al.,
2016; Xu et al., 2015). Panel (a) shows the time series of both Isoprene OA and sulfate at
Centreville site during SOAS. Panel (b) and (c) shows the correlation plot between Isoprene OA
and sulfate from both measurements and model results at two sites (Centreville and Little Rock)
during SOAS.

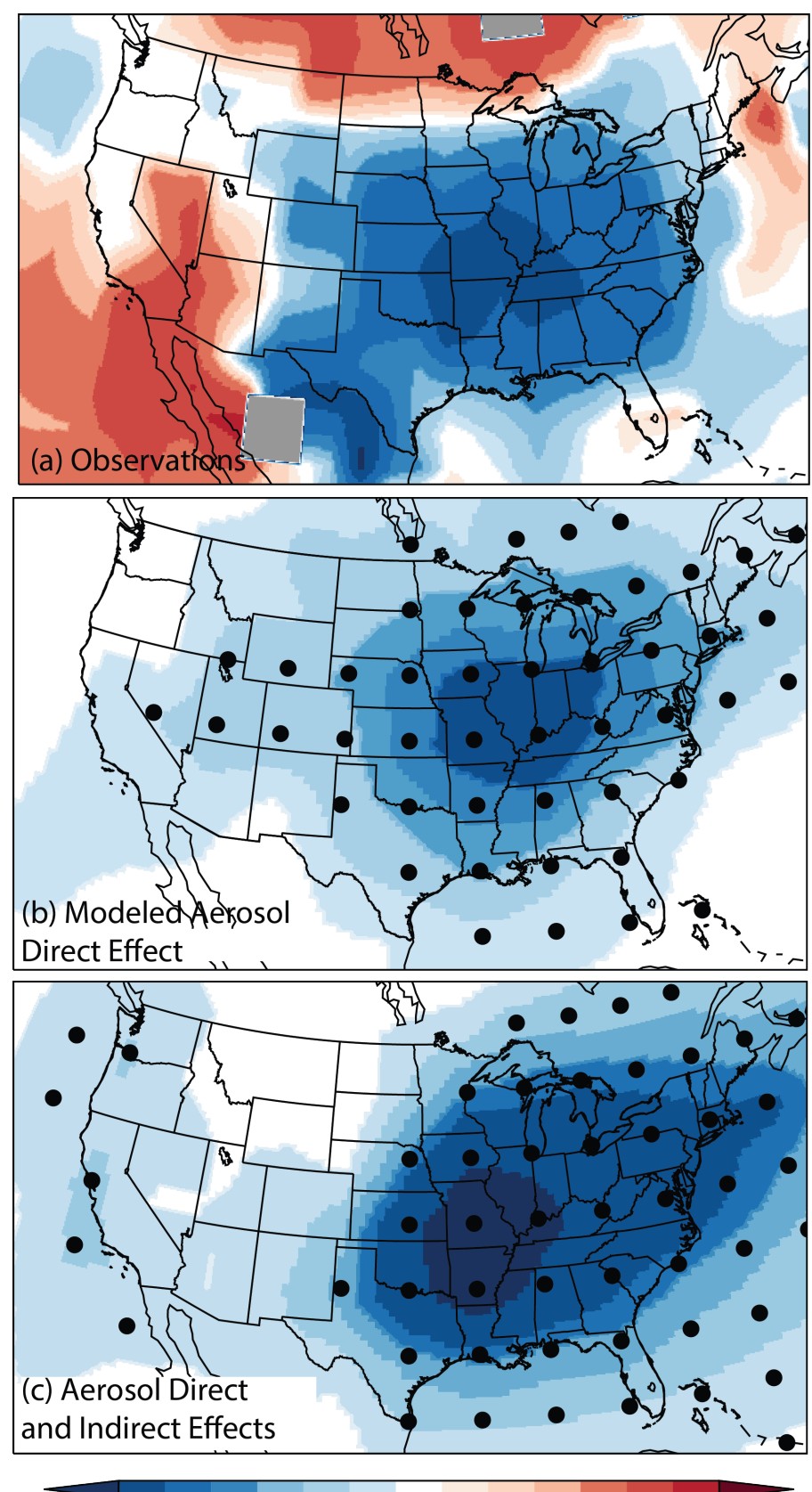

(a) Observations

(b) Modeled Aerosol
Direct Effect

(c) Aerosol Direct
and Indirect Effects

-1.00 -0.75 -0.50 -0.30 -0.20 -0.10 -0.05  0.05  0.10  0.20  0.30  0.50  0.75  1.00

Figure 3 Observed difference in surface air temperature between 1930 and 1990 (a) and modeled
effect of US anthropogenic aerosol sources on surface air temperatures for the 1970–1990 period
when US aerosol loading was at its peak (b and c) (Leibensperger et al., 2012a). Observations are
from      the      NASA      GISS      Surface      Temperature      Analysis      (GISTEMP;
http://data.giss.nasa.gov/gistemp/).    Model values represent the mean difference between 5-
member ensemble GCM simulations including vs. excluding US anthropogenic aerosol sources,
and considering the aerosol direct only (b) and the sum of direct and indirect effects (c). In (b) and
(c), dots indicate differences significant at the 95th percentile.