# Peer review of "Southeast Atmosphere Studies: learning from model-observation syntheses"

_Atmospheric Chemistry and Physics, 2016_

## Referee Comment (RC1) · Anonymous Referee #1 · 11 Mar 2017

This manuscript provides a thorough review of the current knowledge of the atmospheric chemistry in the Southeastern US and the recommendations for future modeling work. My suggestions are mainly associated to the gas-phase chemistry and the modeling approach for gas and particles, which need large improvements in terms of writing. Explanations are needed and results should be clearly presented with proper context (see the specific comments). Although modeling papers are cited, the results are only briefly mentioned. It is still unclear to me after several readings what the current models predict for SE US for gas-phase species and for aerosols, and how do the models compare to the observations. More importantly, what approaches have been adapted in the models and which parameter may lead the largest uncertainties which can be improved based on SAS findings. This should be improved in the revised version. There are also noticeable shifts in writing style between sections possibly be-

cause of the different contributions from the coauthors, which could be improved in the revised version to avoid distracting. I recommend the manuscript be published after a major revision.

Specific comments:

(1) Line 99-100: Please clarify if the interference lead greater or lower concentrations of HOx and which instruments\what study may be affected by this interference.

(2) Line 113-114: Do "Different treatments of this reaction" mean different reaction rates, products, or mechanisms? And quantitatively, how different are the ozone budgets predicted by different models?

(3) Line 133: The conclusion that "isoprene+NO3 reaction is both a major pathway for isoprene removal and for NOx removal" may be right for nighttime. But during the day, photochemistry dominates. Please clarify.

(4) Line 145-146: "many instruments" basically mean GC and PTR-MS (the two commonly used methods for quantifying those products), right? Please be precise.

(5) Line 149-156: It is not obvious for readers about the relationship between the HOx regime and the NOx concentrations. What are the NOx concentrations in the past and now? Why were the past experiments dominated by NOx chemistry but not now? Overall, I think this paragraph fits better with the background section not the findings.

(6) Line 157-166: These two paragraphs seem just restating what was presented in the introduction and background, which do not provide much information about the findings.

(7) Line 168-170: If I understand correctly, SAS has many sites and the HOx data reported by Feiner et al. (2016) are from just one site. For the statement here, the authors need to prove such HOx measurements can represent the whole case of southeast US.

(8) Line 174: "There are many observations that are central to improved understanding

of the detailed mechanisms..." What are the results/key findings then? Please give a summary.

(9) Line 175-176: Please specify what "many of the instruments used in this experiment and in many prior ones" stands for. References are needed.

(10) Line 183-186: Please explain how the lifetime of organic nitrates could affect "the lifetime of NOx, the spatial pattern of transported Nox, and the oxidation rates by OH, O3, and NO3".

(11) Line 189-190: I am curious about the satellite findings of the ratio of glyoxal to HCHO. Does it different or similar to 2% which was observed in SAS? Please clarify.

(12) Line 191-203: Paragraphs (5) and (7) are repeated statements in Section 3 – Organic aerosol and do not show much about gas-phase chemistry at least from what was written right now.

(13) Line 208-209: I think that the statements of "no evidence from these studies" (it was one study) and "at any NOx concentration sampled in the rural Southeast" are overstated. What about urban plumes and different seasons? Please clarify.

(14) Line 220-224: There is lack of explanation about "The largest NOx and BVOC emissions are not collocated." Is this also a key finding from SAS? If so, please provide more information and references for it. Besides, I don't think readers can understand the following sentences: "Resolution is especially important for the 15% or so at the tails of the NOx and HCHO distribution – less so for O3...resolve this last 15% which probably requires a horizontal resolution of order 12 km or less". Please clarify.

(15) Line 229-230: The statement of "these errors are approximately linear" needs explanation.

(16) Line 237-238: Is this a common finding for Southeast US?

(17) Line 270-292: Models have difficulties to reproduce the mass loading of OA but the

problems mainly happen in urban area and aloft. It would be very helpful to summarize the model results regarding Southeast US here.

(18) Line 298: What are "a diversity of modeling approaches"?

(19) Line 313-315: This sentence is confusing. Does "Their structure" mean the structure indicated by the instrument? So "Their structure" is not the actual structures of the original molecules because of the thermal decomposition?

(20) Line 319-322: The authors need to tell readers what the relative humidity and degree of oxygenation of organic compounds in the Southeast US and the conditions leading a phase separation are? The phase behavior also depends on the OA type (precursor).

(21) Line 333-334: "directly equal" is not the right word. Although HOA has been widely used as a surrogate for POA, it has involved some degree of oxidation (if you look at the O:C ratio of HOA in various studies). AMS-PMF analysis also identified factors like CCOA (coal combustion OA), which is in model supposed to be part of POA but it involves some degree of oxidation. Both the semivolatile feature of "POA" and the aging process complicate the model tracers. I would suggest not to link the model tracers to PMF factors. Instead, we need to make cautions that the model OA tracers may not physically match their names for historical reasons. When making comparisons to AMS PMF factors, efforts are needed for understanding the attributions.

(22) Line 354-356: This also argues the statement in Line 133 (also see my comment #3).

(23) Line 373-374: What about the "old" isoprene SOA in the models? Does it overlap somewhat with the IEPOX-SOA? How should the models do?

(24) Line 383-385: The phase also regulates the particle-phase reactions that produce IEPOX-SOA (Kuwata et al., 2015). It is worth to add that.

(25) Line 422: What kind of measurements? Please clarify.

(26) Line 507-513: Since there are different versions of BEIS and MEGAN, there should be first a comparison among versions. So we know for example, in Line 510, which is compared to which.

(27) Line 811-812: This statement is ambiguous. From my reading of section 3.2.6, models cannot neglect SOA from urban emissions for SE US". The parameterization based on CO is an option. But it is unclear at least in section 3.2.6 whether the parameterization based on CO works well for SE US.

Technical remarks: Line 72-74: According to ACP's guidelines, works "submitted to", "in preparation", "in review" should be included in the reference list. Line 74: Warneke et al. is already published. Please cite the AMT version in the reference list. Line 86-87: Add "e.g." in the parentheses. Line 106: Remove the redundant "Peeters et al.,". Line 131: Remove "," after "2012)". Line 137: Since this is a review article and most of the presented results are published ones, I suggest to remove "preliminary" or to rewrite the subtitles. Line 147-148: The last sentence seems being misplaced. Line 154: I could guess that "These experiments" stand for SAS studies. But there is lack of context. Line 217: Missing a word between "there" and "be". Line 288: Remove the "," before "and glyoxal". Line 323-324: This is already stated in Line 291 and since it is about cloud processing which doesn't match directly with the partitioning and phase problems. I would suggest to remove this statement here. Line 322: Pye et al. is already published. Please cite the right one. Line 333-334: Add "the" before "model POA" and "the" before "AMS". Line 461: Should "highest" be "high"? Line 473: "." is missing before "Past". Line 491: $0.25 \times 0.3125$ or $0.25 \times 0.25$? Line 513, 522, 548: According to ACP's guidelines, works "submitted to", "in preparation", "in review" should be included in the reference list. Line 610: "Wm-2" should be "W m-2" Line 623: "a-1" should be "a-1"

Reference: Kuwata, M., Liu, Y., McKinney, K. A., and Martin, S. T.: Physical state and acidity of inorganic sulfate can regulate the production of secondary organic material from isoprene photooxidation products, Phys. Chem. Chem. Phys., 17, 5670-5678,

10.1039/C4CP04942J, 2015.

---

## Referee Comment (RC2) · Anonymous Referee #2 · 14 Mar 2017

This paper discusses recent results and open questions regarding air quality in the southeast U.S. It summarizes the first analyses of the Summer 2013 field campaigns and outlines open questions. It provides recommendations for directions and methods of future analyses of these campaigns.

I think this paper would benefit greatly if it were re-written as a review of our current understanding of the Southeast Atmosphere and less as a report of the workshop in 2015. The workshop could certainly be mentioned in the introduction, but it seems unnecessary to mention it in the abstract and elsewhere in the paper. A workshop summary does not seem appropriate for a journal article in ACP, but a review of results and analyses, and guidance for future research, certainly is. To make it more generally accessible, it would greatly benefit from a more explicit description of all the field campaigns and measurement sites. Thus, in addition to the specific comments below, I feel

[Figure]

a more general re-write of the paper is necessary to make it useful to the community beyond those who participated in SAS.

Specific, technical comments:

The subtitle, "learning from model-observation syntheses", seems a bit awkward. The author list lacks any representation from the NCAR C-130 NOMADSS experiment. Even if the PIs of NOMADSS (i.e., Alex Guenther) were unable to attend the workshop they should be invited to contribute to this paper, which serves as an overview of the Southeast Atmosphere Studies consortium of field campaigns.

A few more figures would be valuable. For example, illustrating some of the findings discussed in Section 4.2 and only referenced with "in preparation" papers. Also a figure in the Introduction showing the flight tracks of the aircraft campaigns and locations of measurement sites would be useful.

l.102: "the HO2+RO2 reaction" - RO2 is not a single compound, so this is not a single reaction.

l.623: "a-1" - missing superscripts

l.625: The MODIS instrument is onboard the Aqua and Terra satellites.

l.814: misplaced comma: . . .lower, and those from MEGAN generally higher, than. . .

---

## Author Comment (AC1) · 26 Oct 2017

We thank two reviewers for their constructive comments. Our responses to the comments are provided below, with the reviewer's comments italicized.

**Reviewer 1**

*This manuscript provides a thorough review of the current knowledge of the atmospheric chemistry in the Southeastern US and the recommendations for future modeling work. My suggestions are mainly associated to the gas-phase chemistry and the modeling approach for gas and particles, which need large improvements in terms of writing. Explanations are needed and results should be clearly presented with proper context (see the specific comments). Although modeling papers are cited, the results are only briefly mentioned. It is still unclear to me after several readings what the current models predict for SE US for gas-phase species and for aerosols, and how do the models compare to the observations. More importantly, what approaches have been adapted in the models and which parameter may lead the largest uncertainties which can be improved based on SAS findings. This should be improved in the revised version. There are also noticeable shifts in writing style between sections possibly because of the different contributions from the coauthors, which could be improved in the revised version to avoid distracting. I recommend the manuscript be published after a major revision.*
Response: The manuscript has been revised substantially to reflect these comments. We now add Table 1 to summarize model evaluations of SAS observations.

Specific comments:
*(1) Line 99-100: Please clarify if the interference lead greater or lower concentrations of HOx and which instruments/what study may be affected by this interference.*
Response: We now revise the text to:
"On the other hand, an interference has been discovered to affect some of these OH instruments (Mao et al., 2012; Novelli et al., 2014; Feiner et al., 2016)."

*(2) Line 113-114: Do "Different treatments of this reaction" mean different reaction rates, products, or mechanisms? And quantitatively, how different are the ozone budgets predicted by different models?*
Response: We now revise the text as:
"Different treatments of these reactions in models including the yield and subsequent fate of daytime isoprene nitrates, cause as much as 20% variations in global ozone production rate and ozone burden among different models (Ito et al., 2009; Horowitz et al., 2007; Perring et al., 2009; Wu et al., 2007; Fiore et al., 2005; Paulot et al., 2012). Large variations mainly stem from different yield of isoprene nitrates (Wu et al., 2007) and the $NO_x$ recycling ratio of these isoprene nitrates (Ito et al., 2009; Paulot et al., 2012)."

*(3) Line 133: The conclusion that "isoprene+NO3 reaction is both a major pathway for isoprene removal and for NOx removal" may be right for nighttime. But during the day, photochemistry dominates. Please clarify.*
Response: We now revise the text as:
"…the reaction is thus a major pathway for nighttime $NO_x$ removal."

*(4) Line 145-146: "many instruments" basically mean GC and PTR-MS (the two commonly used methods for quantifying those products), right? Please be precise.*
Response: We now revise the text as:
"For the case of the major daughter products methylvinylketone (MVK) and methacrolein

(MACR), lab experiments have confirmed that ambient measurements reported to be MVK and MACR, by instruments with metal inlets including gas chromatography (GC) and proton transfer reaction–mass spectrometry (PTR-MS), are more accurately thought of as a sum of MVK, MACR and isoprene hydroperoxides that react on metal and are converted to MVK and MACR (Rivera‑Rios et al., 2014; Liu et al., 2013)."

*(5) Line 149-156: It is not obvious for readers about the relationship between the HOx regime and the NOx concentrations. What are the NOx concentrations in the past and now? Why were the past experiments dominated by NOx chemistry but not now?*
*Overall, I think this paragraph fits better with the background section not the findings.*
**Response**: We have revised and moved this paragraph to the introduction section.

*(6) Line 157-166: These two paragraphs seem just restating what was presented in the introduction and background, which do not provide much information about the findings.*
**Response**: We have removed these two paragraphs.

*(7) Line 168-170: If I understand correctly, SAS has many sites and the HOx data reported by Feiner et al. (2016) are from just one site. For the statement here, the authors need to prove such HOx measurements can represent the whole case of southeast US.*
**Response**: We have revised the paragraph as:

"Radical production: Combining traditional laser-induced fluorescence with a chemical removal method that mitigates potential OH measurement artifacts, Feiner et al. (2016) found that their tower-based measurements of OH and $HO_2$ during SOAS show no evidence for dramatically higher OH than current chemistry predicts in an environment with high BVOCs and low NOx. Instead, they are consistent with the most up-to-date isoprene chemical mechanism. Romer et al. (2016) found that the lifetime of $NO_x$ was consistent with these OH observations and that the major source of $HNO_3$ was isoprene nitrate hydrolysis. Their conclusions would be inconsistent with dramatically higher OH levels, which would imply much more rapid isoprene nitrate production than observed. Other ratios of parent and daughter molecules and chemical lifetimes are also sensitive to OH and these should be explored for additional confirmation or refutation of ideas about OH production at low $NO_x$.

Isoprene vertical flux divergence in the atmospheric boundary layer over the SOAS site and similar forest locations was quantified by Kaser et al. (2015) during the NSF/NCAR C-130 aircraft flights and used to estimate daytime boundary layer average OH concentrations of 2.8 to $6.6 \times 10^6$ molecules $cm^{-3}$. These values, which are based on chemical budget closure, agree to within 20% of directly-observed OH on the same aircraft. After accounting for the impact of chemical segregation, Kaser et al. (2015) found that current chemistry schemes can adequately predict OH concentrations in high isoprene regimes. This is also consistent with the comparison between measured and modeled OH reactivity on a ground site during SOAS, which show excellent agreement above the canopy of an isoprene-dominated forest (Kaiser et al., 2016). "

*(8) Line 174: "There are many observations that are central to improved understanding of the detailed mechanisms: : :" What are the results/key findings then? Please give a summary.*
**Response**: We have now revised this paragraph as :
"These experiments have been guided and/or corroborated by analyses of field observations of total and speciated alkyl nitrates (Romer et al., 2016; Nguyen et al., 2015; Xiong et al., 2015; Lee

et al., 2016), IEPOX/ISOPOOH (Nguyen et al., 2015), glyoxal (Min et al., 2016), HCHO (Wolfe et al., 2016), OH reactivity (Kaiser et al., 2016), and airborne fluxes (Wolfe et al., 2015). Recent modeling studies have incorporated these mechanisms to some extent and showed success on reproducing temporal and spatial variations of these compounds (Su et al., 2016; Fisher et al., 2016; Travis et al., 2016; Zhu et al., 2016; Li et al., 2017; Li et al., 2016), as summarized in Table 1."

*(9) Line 175-176: Please specify what "many of the instruments used in this experiment and in many prior ones" stands for. References are needed.*
**Response**: We now removed this sentence.

*(10) Line 183-186: Please explain how the lifetime of organic nitrates could affect "the lifetime of NOx, the spatial pattern of transported NOx, and the oxidation rates by OH, O3, and NO3".*
**Response**: We now revised as:
"The assumed lifetime and subsequent fate of organic nitrates can profoundly influence $NO_x$ levels across urban-rural gradients (Browne and Cohen, 2012; Mao et al., 2013), affecting oxidant levels and formation of secondary organic aerosol (SOA)."

*(11) Line 189-190: I am curious about the satellite findings of the ratio of glyoxal to HCHO. Does it different or similar to 2% which was observed in SAS? Please clarify.*
**Response**: We now revised as:
"Widespread ambient confirmation of the ratio is difficult because of large biases in satellite glyoxal quantification (Chan Miller et al., 2017)."

*(12) Line 191-203: Paragraphs (5) and (7) are repeated statements in Section 3 – Organic aerosol and do not show much about gas-phase chemistry at least from what was written right now.*
**Response**: We have merged these paragraphs into Section 3 and 4.

*(13) Line 208-209: I think that the statements of "no evidence from these studies" (it was one study) and "at any NOx concentration sampled in the rural Southeast" are overstated. What about urban plumes and different seasons? Please clarify.*
**Response**: We now revised as:
"Measurements and modeling effort on OH show no indication of a need for empirical tuning factors to represent OH chemistry in the rural Southeast US."
Discussion of OH in other scenarios is beyond the scope of this work.

*(14) Line 220-224: There is lack of explanation about "The largest NOx and BVOC emissions are not collocated." Is this also a key finding from SAS? If so, please provide more information and references for it. Besides, I don't think readers can understand the following sentences: "Resolution is especially important for the 15% or so at the tails of the NOx and HCHO distribution – less so for O3: : :resolve this last 15% which probably requires a horizontal resolution of order 12 km or less". Please clarify.*
**Response**: We now revised as:
"The largest $NO_x$ and BVOC emissions are not collocated, as one is mainly from mobile sources and power plants and the other one is mainly from forests (Yu et al., 2016; Travis et al., 2016). As a result, model resolution can impact predicted concentrations of trace species. Different model resolutions may lead to as much as 15% differences at the tails of the $NO_x$ and HCHO distribution—less so for $O_3$ (Yu et al., 2016; Valin et al., 2016)."

*(15) Line 229-230: The statement of "these errors are approximately linear" needs explanation.*

**Response**: It is stated as:

"At the low $NO_x$ characteristic of the Southeast U.S. these errors are approximately linear—that is, a 15% error in $NO_x$ should correspond to a 15% error in OH, isoprene and other related species."

*(16) Line 237-238: Is this a common finding for Southeast US?*
**Response**: We have expanded this paragraph:

"A significant fraction of isoprene remains at sunset and is available for oxidation via $O_3$ or $NO_3$ at night.  Analysis of nighttime isoprene and its oxidation products in the residual layer in the northeast U.S. in 2004 suggested this fraction to be 20% (Brown et al. 2009).  Preliminary analysis from SENEX suggested a similar fraction, although the analysis depends on the emission inventory for isoprene, and would be 10-12% if isoprene emissions were computed from MEGAN (see Section 4.2 for the difference between BEIS and MEGAN).  This fact might be a useful diagnostic of boundary layer dynamics and nighttime chemistry in models. The overnight fate of this isoprene depends strongly on available $NO_x$ (see above).  More exploration of the model prediction of the products of $NO_3$ + isoprene and additional observations of those molecules will provide insight into best practices for using it as a diagnostic of specific model processes."

*(17) Line 270-292: Models have difficulties to reproduce the mass loading of OA but the problems mainly happen in urban area and aloft. It would be very helpful to summarize the model results regarding Southeast US here.*
**Response**: We have added Table 1 to summarize model results regarding Southeast US. And we also add:
"Models have difficulties to reproduce the mass loading of OA in both urban and rural areas, although order-of-magnitude underestimates have only been observed consistently for urban pollution (e.g. Volkamer et al., 2006; Hayes et al., 2015). For example, CMAQ underestimates OA by 17% at SEARCH network sites with higher overestimates and underestimates at night and during the day respectively (Pye et al., 2017a). Furthermore, current OA algorithms often rely on highly parameterized empirical fits to laboratory data that may not capture the role of oxidant (OH vs $O_3$ vs $NO_3$) or peroxy radical fate. The peroxy radical fate for historical experiments in particular, may be biased compared to the ambient atmosphere where peroxy radical lifetimes are longer and autoxidation can be important."

*(18) Line 298: What are "a diversity of modeling approaches"?*
**Response**: We now revise as:
"A diversity of modeling approaches, including direct scaling with emissions, reactive uptake of gaseous species, and gas-aerosol partitioning etc., is encouraged to provide insight into OA processes, while trying to make use of all available experimental constraints to evaluate the models."

*(19) Line 313-315: This sentence is confusing. Does "Their structure" mean the structure indicated by the instrument? So "Their structure" is not the actual structures of the original molecules because of the thermal decomposition?*
**Response**: We now revise as:
"In some instances (e.g. for key IEPOX-SOA species), observations indicate that detected OA species are significantly less volatile than their structure indicates, likely due to thermal decomposition of their accretion products or inorganic-organic adducts in instruments (Lopez-Hilfiker et al., 2016; Hu et al., 2016; Isaacman-VanWertz et al., 2016; Stark et al., 2017)."

*(20) Line 319-322: The authors need to tell readers what the relative humidity and degree of oxygenation of organic compounds in the Southeast US and the conditions leading a phase separation are? The phase behavior also depends on the OA type (precursor).*
**Response**: We now revise as:
"However, due to the high relative humidity (average RH is 74%, see Weber et al. (2016)) and degree of oxygenation of organic compounds (OM/OC is 1.9-2.25, see below) in the southeast US atmosphere, inorganic-rich and organic-rich phases may not be distinct (You et al., 2013) and more advanced partitioning algorithms accounting for a mixed inorganic-organic-water phase may be needed (Pye et al., 2017a; Pye et al., 2017b)."

*(21) Line 333-334: "directly equal" is not the right word. Although HOA has been widely used as a surrogate for POA, it has involved some degree of oxidation (if you look at the O:C ratio of HOA in various studies). AMS-PMF analysis also identified factors like CCOA (coal combustion OA), which is in model supposed to be part of POA but it involves some degree of oxidation. Both the semivolatile feature of "POA" and the aging process complicate the model tracers. I would suggest not to link the model tracers to PMF factors. Instead, we need to make cautions that the model OA tracers may not physically match their names for historical reasons. When making comparisons to AMS PMF factors, efforts are needed for understanding the attributions.*
**Response**: We now revise as:
"Thus care should be exercised in evaluating model species such as POA with Aerosol Mass Spectrometer (AMS) Positive Matrix Factorization (PMF) factors such as hydrocarbon-like OA (HOA)."

*(22) Line 354-356: This also argues the statement in Line 133 (also see my comment #3).*
**Response**: We have added this in the text:
"Initial studies indicate that monoterpene oxidation can be an important sink of $NO_x$ and an important source of aerosol precursors (Lee et al., 2016; Ayres et al., 2015)."

*(23) Line 373-374: What about the "old" isoprene SOA in the models? Does it overlap somewhat with the IEPOX-SOA? How should the models do?*
**Response**: It appears the the old isoprene SOA contributes much less than IEPOX-SOA, according to recent study. We now add:
"D'Ambro et al. (2017) predicts IEPOX will be the major precursor to SOA under low-$NO_x$ conditions when peroxy radical lifetimes are atmospherically relevant, which has not always been the case in older experiments."

*(24) Line 383-385: The phase also regulates the particle-phase reactions that produce IEPOX-SOA (Kuwata et al., 2015). It is worth to add that.*
**Response:** We now revise as:
"Current pathways for IEPOX-SOA formation (Eddingsaas et al., 2010) involve acidity in aqueous solutions (Kuwata et al., 2015), but several studies suggest that IEPOX-SOA is not correlated well with aerosol acidity or aerosol water (Budisulistiorini et al., 2017; Xu et al., 2015)."

*(25) Line 422: What kind of measurements? Please clarify.*
**Response:** We now revise as:
"On the other hand, SEARCH measurements agree well with research community instruments in Centerville site, such as AMS."

*(26) Line 507-513: Since there are different versions of BEIS and MEGAN, there should be first a*

*comparison among versions. So we know for example, in Line 510, which is compared to which.*

**Response:** We now add a new paragraph to elaborate on the difference between MEGAN and BEIS:

"A simplification used in current biogenic emission models including BEIS3.13, BEIS3.6, and MEGAN2.1 is that all high isoprene emitting species are assigned the same isoprene emission factor. For example, all North American species of Quercus (oak), Liquidambar (sweetgum), Nyssa (tupelo), Platanus (sycamore), Salix (willow), Robinia (locust) and Populus (poplar and aspen) are assigned a single value based on the average of an extensive set of enclosure measurements conducted in North Carolina, California and Oregon in the 1990s (Geron et al., 2001). Earlier studies had reported isoprene emission factors for these tree species that ranged over more than an order of magnitude (Benjamin et al., 1996). Geron et al. (2001) showed that by following specific measurement protocols, including leaf cuvettes with environmental controls and ancillary physiological measurements such as photosynthesis, the variability dropped from over an order of magnitude to about a factor of 3. They concluded that this remaining variability was due at least as much to growth conditions as to species differences and so recommended that a single isoprene emission factor be used for all of these species. Recent aircraft flux measurements (Misztal et al., 2016; Yu et al., 2017) indicate that there is at least a factor of two difference in the isoprene emission factors of these species. This could be due to a genetic difference in emission capacity and/or differences in canopy structure. The aircraft measurements indicate that sweetgum and tupelo emission factors are similar to the value used in BESI3.13 and BEIS3.6 while the California oak emission factor is similar to that used in MEGAN2.1. The aircraft based estimate of southeastern oak emission factors falls between the BEIS3.6 and MEGAN2.1 values. As a result, aircraft flux measurements in the southeastern US are higher than BEIS3.13/BEIS3.6 and lower than MEGAN2.1. The MEGAN3 emission factor processor provides an approach for synthesizing available emission factor data and can be used to account for the emission rate variability observed by these aircraft flux studies (Guenther et al., in preparation)."

*(27) Line 811-812: This statement is ambiguous. From my reading of section 3.2.6, models cannot neglect SOA from urban emissions for SE US". The parameterization based on CO is an option. But it is unclear at least in section 3.2.6 whether the parameterization based on CO works well for SE US.*

**Response:** We now revise as:

"A simple parameterization based on CO emissions (Hayes et al., 2015) may be adequate for incorporating this source in modeling studies and has shown good results for the Southeast US (Kim et al., 2015), but care should be taken to evaluate the CO emissions when using it."

*Technical remarks:*
*Line 72-74: According to ACP's guidelines, works "submitted to","in preparation", "in review" should be included in the reference list. Line 74: Warneke et al. is already published. Please cite the AMT version in the reference list.*
**Response:** Corrected.

*Line 86-87: Add "e.g." in the parentheses.*
**Response:** Added.

*Line 106: Remove the redundant "Peeters et al.,".*
**Response:** Corrected.

*Line 131: Remove "," after "2012)".*

**Response:** Corrected.

*Line 137: Since this is a review article and most of the presented results are published ones, I suggest to remove "preliminary" or to rewrite the subtitles.*
**Response:** We now revise to "Major relevant findings".

*Line 147-148: The last sentence seems being misplaced.*
**Response:** Corrected. We add a new paragraph on monoterpene chemistry:
"Higher-order terpenes: Monoterpene and sesquiterpene chemistry requires continued investigation. Initial studies indicate that monoterpene oxidation can be an important sink of $NO_x$ and an important source of aerosol precursors (Lee et al., 2016; Ayres et al., 2015). Additional analysis is needed to understand the role of monoterpenes. We note that because our understanding of isoprene chemistry has been changing so rapidly and because the role of isoprene sets the stage for evaluating the role of monoterpenes, we are now in a much better position to evaluate the role of monoterpene chemistry."

*Line 154: I could guess that "These experiments" stand for SAS studies. But there is lack of context.*
**Response:** This paragraph has been merged into introduction section.

*Line 217: Missing a word between "there" and "be".*
**Response:** fixed.

*Line 288: Remove the "," before "and glyoxal".*
**Response:** fixed.

*Line 323-324: This is already stated in Line 291 and since it is about cloud processing which doesn't match directly with the partitioning and phase problems. I would suggest to remove this statement here.*
**Response:** We have created a new subsection for Cloud SOA.

*Line 322: Pye et al. is already published. Please cite the right one.*
**Response:** fixed.

*Line 333-334: Add "the" before "model POA" and "the" before "AMS".*
**Response:** fixed.

*Line 461: Should "highest" be "high"?*
**Response:** fixed.

*Line 473: "." is missing before "Past".*
**Response:** fixed.

*Line 491: 0.25x0.3125 or 0.25x 0.25?*
**Response:** fixed.

*Line 513, 522, 548: According to ACP's guidelines, works "submitted to", "in preparation", "in review" should be included in the reference list.*
**Response:** fixed

*Line 610: "Wm-2" should be "W m-2"*
**Response:** fixed.

*Line 623: "a-1" should be "a-1"*
**Response:** fixed.

**Reviewer 2**

*This paper discusses recent results and open questions regarding air quality in the southeast U.S. It summarizes the first analyses of the Summer 2013 field campaigns and outlines open questions. It provides recommendations for directions and methods of future analyses of these campaigns. I think this paper would benefit greatly if it were re-written as a review of our current understanding of the Southeast Atmosphere and less as a report of the workshop in 2015. The workshop could certainly be mentioned in the introduction, but it seems unnecessary to mention it in the abstract and elsewhere in the paper. A workshop summary does not seem appropriate for a journal article in ACP, but a review of results and analyses, and guidance for future research, certainly is. To make it more generally accessible, it would greatly benefit from a more explicit description of all the field campaigns and measurement sites. Thus, in addition to the specific comments below, I feel a more general re-write of the paper is necessary to make it useful to the community beyond those who participated in SAS.*
**Response:**
We have now largely revised the text as a review of results and analyses of the Southeast Atmosphere Studies, with recommendations for future modeling effort. The content has also been expanded to include many recent results.

*Specific, technical comments:*
*The subtitle, "learning from model-observation syntheses", seems a bit awkward. The author list lacks any representation from the NCAR C-130 NOMADSS experiment. Even if the PIs of NOMADSS (i.e., Alex Guenther) were unable to attend the workshop they should be invited to contribute to this paper, which serves as an overview of the Southeast Atmosphere Studies consortium of field campaigns.*
**Response:** We have now invited Dr. Alex Guenther, Dr. Steve Brown and a number of PIs to contribute to this paper. The author list has been largely expanded.

*A few more figures would be valuable. For example, illustrating some of the findings discussed in Section 4.2 and only referenced with "in preparation" papers. Also a figure in the Introduction showing the flight tracks of the aircraft campaigns and locations of measurement sites would be useful.*
**Response:** Section 4.2 is now largely expanded to elaborate on recent findings. Flight tracks of the aircraft campaigns and locations of measurement sites are covered in Carlton et al. (Synthesis of the Southeast Atmosphere Studies: investigating fundamental atmospheric chemistry questions, BAMS, 2017, accepted). This paper is intended to to serve as a guidance for future modeling efforts.

*l.102: "the HO2+RO2 reaction" - RO2 is not a single compound, so this is not a single reaction.*
**Response:** fixed.

*l.623: "a-1" - missing superscripts*
**Response:** fixed.

*l.625: The MODIS instrument is onboard the Aqua and Terra satellites.*
**Response:** fixed.

*l.814: misplaced comma: : : :lower, and those from MEGAN generally higher, than: : :*
**Response:** fixed.

Reference

[revised manuscript text omitted]

---

## Author Response (AR2)

We thank reviewer 3 and editor for their constructive comments. Our responses to the comments are provided below, with the reviewer's comments italicized.

**Reviewer 3**

*The authors presented a brief review of the findings on the model-observation syntheses based on SAS which is a useful and big effort to summarize comprehensive achievements from a series of campaigns performed in Southeast US. For the gas phase chemistry part, the major findings, the recommendations and key model diagnostics seems a bit lengthy and with some redundancies in the part of background, findings, and model recommendations, etc. I recommend publication after the authors to address the following comments.*

*Specific comments:*
*-The summarized major findings are kind of well-known. And the authors may provide arguments from two sides. One is the specific feature of the SAS and the other is the general implications on the global scale.*
**Response**: The global implications of these findings remain largely unclear. We decide to leave this discussion to future studies.

*-The model recommendations and the key model diagnostics are nice but not fully supported by the literatures or by the evidences provided in this paper. For example, the use of the ratios like NO2/HNO3 and MVK/Isoprene shall be with cautious when there is a mixing of air masses with different ages. The recommendation for the isoprene chemistry is clear and strong. But what is the reason for the explicit chemistry through the first and second generation of isoprene oxidation. To achieve better OH model results or that of SOA? NO3 chemistry shall also be important for the NOx removal. The segregation problem in chemical transport model system is well known. Is the 12 km recommendation more applied for southeast US or the authors think that is applicable globally and why is that?*
**Response**: We now state:
"We recommend that there should be explicit chemistry through the first and second generation of isoprene oxidation, to better illustrate the role of isoprene in ozone production, OH budget and SOA production."

We also revise:
"$NO_3$ chemistry is an important element of VOC oxidation, $NO_x$ removal and aerosol production. $NO_3$ chemistry should be included in models that do not explicitly take it into account, both as a loss process of VOCs and $NO_x$ and as a source of aerosols."

We also add:
"We note that this recommendation is only applied for Southeast US, and further studies are warranted to apply this result to other locations."

*-The open questions shall not be only questions. Maybe the authors want to add some comments on these questions as well. Some part of the open questions are out of the context. It is of course important to know the water in aerosols and understand the links between chemical mixing and boundary layer dynamics. But what is the specific study which could indicate or show its importance/uncertainties in the framework of SAS.*

**Response**: We have moved this question to Section 3.4, with relevant studies mentioned in Section 3.2.8.

*Technical comments:*
*L196, "radical production" better changed to "radical simulation"*
**Response**: Done.

**Editor**
*This time one of the previous reviewers (reviewer #2) and a new reviewer (reviewer #1) made comments. Minor revision is necessary based on the comments from the reviewer #1. My comments as follows should also be taken into account:*

*1. The manuscript is improved but became lengthy and conciseness is preferred. I understand that contents were added based on the first comments by the reviewers. But as the nature of the journal review process, it would be better to focus on the original points that the authors made without quite large expansion during the revision. In the background subsections (2.1, 3.1, 4.1, and 5.1), papers published in 2017 are even found. Some of them might be better mentioned only in the findings subsections. The background description on the monoterpene chemistry (lines 211-251 in the track change version) is even longer than the findings in the following section (lines 391-397) and could be shortened.*
**Response**: We have removed several citations published in 2017 from background sections. We have also shortened the background description on monoterpene chemistry (now Lines 170-181).

*2. residual layer (line 371 in the track change version) should be written as RL, as already appeared in line 339. Please check abbreviation thoroughly.*
**Response**: Fixed.

*3. Lee et al. 2016a (line 654, in the track change version) is not found in the literature list. Completeness should be checked also for other cited papers.*
**Response**: Fixed.

[revised manuscript text omitted]